# Single nucleus RNA-sequencing defines unexpected diversity of cholinergic neuron types in the adult mouse spinal cord

Mor R. Alkaslasi [1,2,3], Zoe E. Piccus[1,2,3], Sangeetha Hareendran[1], Hanna Silberberg[1], Li Chen[1], Yajun Zhang[1], Timothy J. Petros [1] & Claire E. Le Pichon [1✉]

In vertebrates, motor control relies on cholinergic neurons in the spinal cord that have been extensively studied over the past hundred years, yet the full heterogeneity of these neurons and their different functional roles in the adult remain to be defined. Here, we develop a targeted single nuclear RNA sequencing approach and use it to identify an array of cholinergic interneurons, visceral and skeletal motor neurons. Our data expose markers for distinguishing these classes of cholinergic neurons and their rich diversity. Specifically, visceral motor neurons, which provide autonomic control, can be divided into more than a dozen transcriptomic classes with anatomically restricted localization along the spinal cord. The complexity of the skeletal motor neurons is also reflected in our analysis with alpha, gamma, and a third subtype, possibly corresponding to the elusive beta motor neurons, clearly distinguished. In combination, our data provide a comprehensive transcriptomic description of this important population of neurons that control many aspects of physiology and movement and encompass the cellular substrates for debilitating degenerative disorders.

[1] Eunice Kennedy Shriver National Institute of Child Health and Human Development, National Institutes of Health, Bethesda, MD, USA. [2] Department of Neuroscience, Brown University, Providence, RI, USA. [3] These authors contributed equally: Mor R. Alkaslasi, Zoe E. Piccus. ✉email: claire.lepichon@nih.gov

Cholinergic spinal cord neurons are essential for all aspects of motor control including voluntary contractions of the limbs and involuntary motions of internal organs. These cholinergic neurons can be divided into three main types: skeletal motor neurons, visceral motor neurons, and interneurons, with distinct functions in motor control[1]. The two types of motor neurons are particularly unusual as their cell bodies are located in the central nervous system and project to the periphery to connect the brain to the body. Skeletal motor neurons (MNs) innervate skeletal muscle to coordinate muscle contraction, drive locomotion, and mediate fine motor control. Pre-ganglionic autonomic neurons, or visceral MNs, project to autonomic ganglion neurons that in turn innervate smooth muscle and glands to control almost all physiological responses and organs of the body. The cholinergic interneurons are critical for the local spinal circuitry including regulating motor neuron excitability[2,3]. Together, these three neuronal classes comprise a relatively small population among all spinal cord cells that communicate to their target cells with the excitatory neurotransmitter acetylcholine. Previous work has shown that each of the three main classes can be divided into subtypes with specific properties and specializations[1]. However, the true diversity of spinal cholinergic neurons remains unknown.

One area of intense focus has been defining subtypes of skeletal MNs since the dysfunction of this class is a major component of several diseases. Clinically, it has become clear that some subtypes are more susceptible than others to degeneration. This is particularly striking in motor neuron diseases such as spinal muscular atrophy and amyotrophic lateral sclerosis (ALS) where select MNs degenerate while others are spared[4–7]. For example, among affected neurons in ALS, skeletal MNs innervating fast-twitch extrafusal fibers degenerate earlier than those innervating slow-twitch fibers[7,8]. Most mutations linked to ALS are in widely expressed genes, yet for unknown reasons MN subtypes are more susceptible to death than other cell types, leading to the idea that cell-intrinsic characteristics determine this vulnerability[7,9]. If so, a transcriptomic definition of skeletal motor neuron subtypes could unveil potential causes of cell-type susceptibility, define better markers for studying degeneration and provide strategies to selectively control gene expression in subsets of MNs.

With the recent advances in sequencing technologies, a few studies have transcriptionally profiled individual spinal cord neurons in development[10,11] and in the adult[12]. But due to the technical approaches adopted, the rarity of cholinergic neurons among all spinal cord cells, as well as the large size of motor neurons, only a few cholinergic neurons have been successfully sequenced[11,12]. Here, we use a genetic strategy to permanently mark cholinergic nuclei in the adult mouse spinal cord and selectively enrich them for single-nucleus RNA sequencing (snRNAseq). This approach allowed us to systematically classify cholinergic neurons and generate an atlas of their transcriptional identities. Our results exposed an array of cholinergic interneuron subtypes and revealed significant molecular diversity amongst skeletal MNs, identifying targets for exploring their function. Most surprisingly, we discovered an extensive diversity of visceral motor neuron subtypes and defined their anatomic organization along the length of the spinal cord. Together, the data (available at www.spinalcordatlas.org) provide a detailed view of spinal cord cholinergic neuron types, identify molecular signatures for each, and provide insights into their normal physiological functions.

## Results
**Transcriptional profiling of adult mouse spinal cholinergic neurons.** One goal of single-cell transcriptomics is to use the full transcriptome to define neuronal classes and identify diagnostic

mRNA marker combinations, reflecting functional or anatomical distinctions between types of neurons. Nuclear RNA sequencing provides an important technical advance for single-cell profiling from tissues such as the spinal cord that are difficult to dissociate into single cells[12,13]. We selected a single nucleus RNA sequencing strategy for this reason, and to ensure the accurate and unbiased profiling of all types of motor neurons, including those of large diameter that is typically not viably captured using single-cell isolation[14]. An added advantage of the single nuclei preparation compared with single cells is that it is very rapid and therefore avoids transcriptional stress responses that can occur during tissue collection and dissociation, providing a more accurate baseline representation of cell classes[11,13,14]. However, even using snRNAseq[12] very few cholinergic neurons were identified, reflecting their sparse representation in the spinal cord.

To selectively enrich for nuclei from spinal cholinergic neurons, we bred mice where cholinergic cells in the adult spinal cord were labeled using Chat-IRES-Cre. Our genetic strategy permanently marks their nuclei with a bright fluorescent protein attached to the nuclear envelope[15] (Fig. 1). Whole tissue immunolabeling and clearing revealed bright nuclei throughout the spinal cord in the expected locations for cholinergic cells— ventral horn, lateral column, and intermediate zone (Supplementary Fig. 3c; Supplementary Movie 1), thus verifying our strategy. In order to be able to spatially map individual cells to different spinal cord regions, we dissected the spinal cords into 3 regions: cervical, thoracic, and lumbo-sacral, and processed them separately (Fig. 1). We next proceeded to isolate nuclei from frozen tissue, harvest GFP+ nuclei via fluorescence-activated cell sorting (FACS) (Supplementary Fig. 1), perform snRNA-Seq (see "Methods" section), and characterize a total of 34,231 single nuclei that met standard criteria (e.g., independent transcripts, number of genes, level of mitochondrial transcripts). Preliminary analysis revealed that the nuclei segregated into 38 distinct clusters (Supplementary Fig. 2a). Common cell type markers demonstrated that, while all of the nuclei were neuronal (Snap25-expressing, Supplementary Fig. 2c), 22 of the clusters were not cholinergic, reflecting excitatory or inhibitory neurons that had expressed Chat-IRES-Cre, either developmentally or due to leaky expression of Cre (Supplementary Fig. 2b–c, 3 a–b). After removal of contaminating non-cholinergic neurons, analysis of the 16,042 Chat-expressing nuclei identified 21 transcriptionally distinct subtypes (Fig. 2a; Supplementary Table 1), exposing considerably greater diversity for this class of neurons than the two groups of cholinergic neurons that had been previously described[11]. However, we did not observe any sex-specific differences among cholinergic neurons (Supplementary Fig. 4).

**Classification of main cholinergic neuron types.** How do the 21 distinct transcriptomic classes of cholinergic neurons relate to the 3 major categories of spinal neurons? At the most basic level, the distribution of published markers for these cells in our data should help define the roles of the diverse clusters. Moreover, skeletal motor neurons, cholinergic interneurons, and visceral motor neurons each reside in distinct locations within the spinal cord (Fig. 1; boxed areas in Fig. 2e–f; Supplementary Fig. 3c)[16–18]. Skeletal motor neurons are exclusively located in the ventral horn, with axons that exit via the ventral roots to innervate and control the skeletal muscles of the body. Cholinergic interneurons, as modulators and regulators of neuronal activity, are small neurons found primarily around the central canal and the intermediate zone[2,19]. Visceral pre-ganglionic motor neurons are localized to the lateral column of the spinal cord and project to the ganglion neurons, which in turn innervate cardiac and smooth muscles for organ control[20,21]. We reasoned that, if molecular markers for

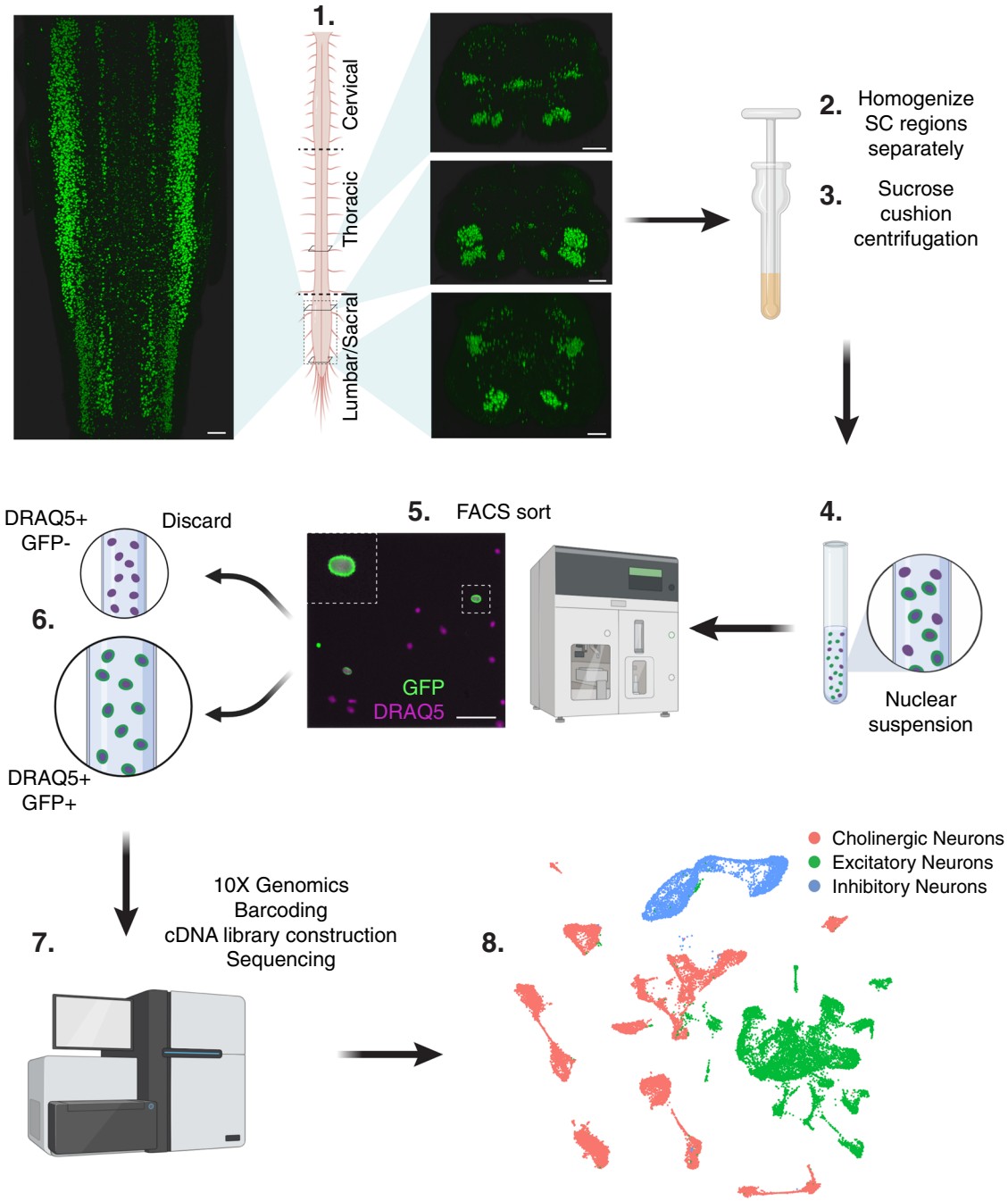

**Fig. 1 Strategy for single nucleus RNA sequencing of spinal cord cholinergic neurons.** Spinal cords were extruded from Chat-IRES-Cre::CAG-Sun1-sfGFP mice and separated into 3 regions (Cervical, Thoracic, and Lumbar/Sacral) that were processed separately (1). Image from a whole cleared spinal cord (SC) immunolabeled for GFP (Supplementary Movie 1). Tissue was lysed (2) and nuclei were isolated by density gradient centrifugation (3). Nuclear suspensions (4) were sorted using fluorescent-activated cell sorting (FACS) (5) to select singlet GFP-positive DRAQ5-positive nuclei (6) that were processed for single nucleus RNA sequencing using the 10X Genomics Chromium platform (7). Single nucleus cDNA libraries were sequenced together on an Illumina HiSeq (7), then analyzed (8). Figure created with BioRender.com. Scale bars, (1) 200 μm, (5) 50 μm.

each subgroup could be identified, their expression pattern in the spinal cord should be diagnostic.

We began by exploring the expression of several molecules previously assigned to subtypes of cholinergic neurons and found these markers mapped to specific clusters of neurons (Fig. 2b). Spinal motor columns are specified during development and can be distinguished by the expression of marker genes in the embryo[1]. However, many of these embryonic markers were either undetected, only lowly expressed, or non-specific to particular

cholinergic subtypes in our dataset, suggesting the expression pattern of most of these genes is not maintained in the adult once development is complete (*Alcam*, *Foxp1*, *Isl1*, *Isl2*, *Lhx3*, *Mnx1*; Supplementary Fig. 14). Nonetheless, some markers identified in development proved useful in our analysis. For example, *Pax2*, a well-established marker for cholinergic interneurons[22], was highly expressed in 7 clusters that we renamed I1-I7. One small cluster, I8, expressed *Mpped2*, a gene shared with all other interneuron clusters (Fig. 2c; Supplementary Fig. 5c) but not the

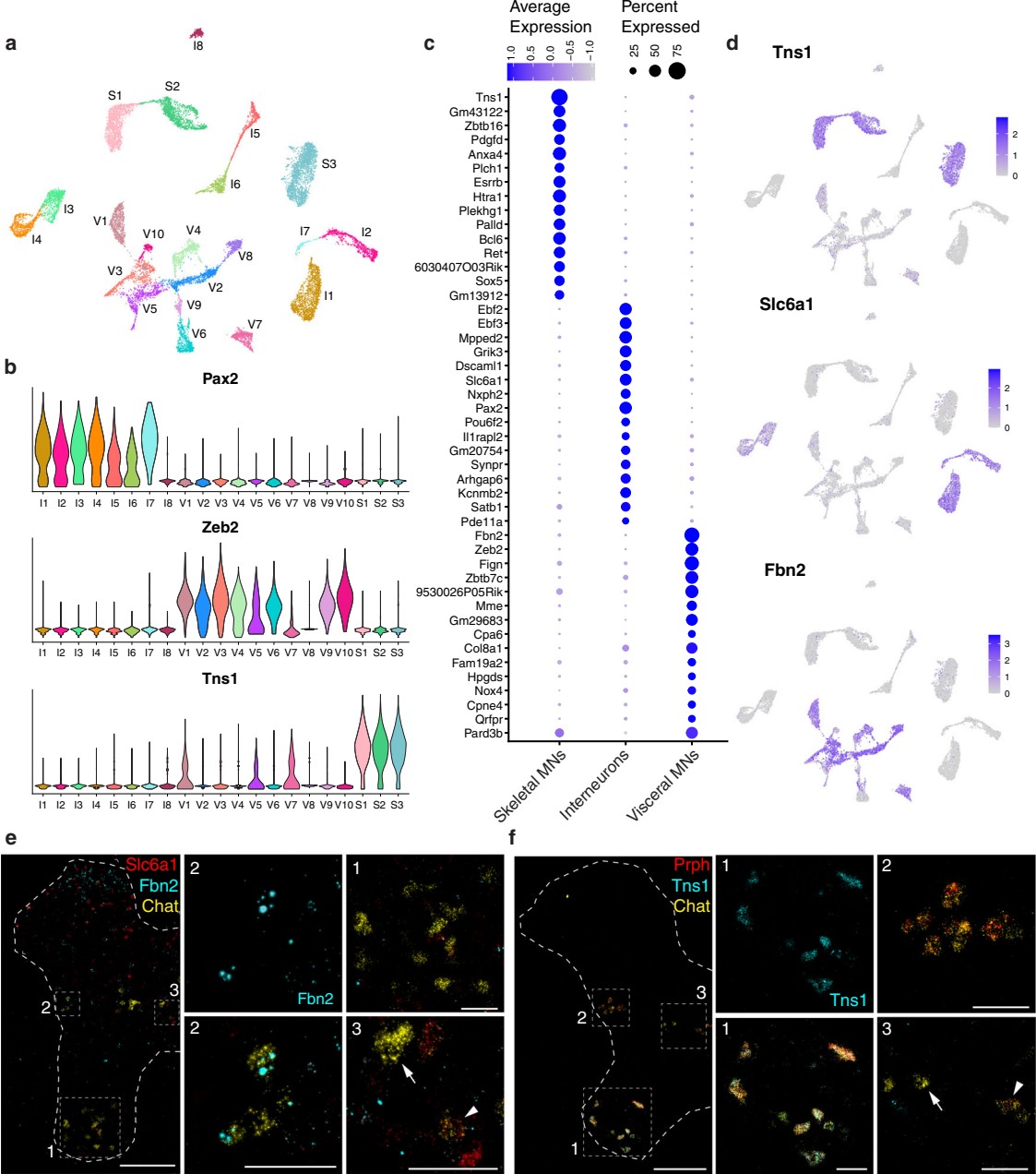

**Fig. 2 Classification of cholinergic neurons reveals the extensive diversity of motor neurons and interneurons. a** UMAP of all 16,042 cholinergic nuclei showing 21 distinct clusters. **b** Expression of marker genes for cholinergic interneurons (*Pax2*), visceral MNs (*Zeb2*), and skeletal MNs (*Tns1*) in the 21 cholinergic clusters. **c** Dot plot showing the top 15 marker genes for each main cholinergic subtype. **d** Feature plots depicting expression of the top novel marker gene for each cholinergic subtype. **e** RNAScope for novel marker genes for a subset of interneurons (*Slc6a1*, red) and visceral MNs (*Fbn2*, cyan) co-expressing with *Chat* (yellow). Interneurons in the intermediate zone (box 3) are double positive for *Slc6a1* and *Chat* (arrowhead) or only express *Chat* (arrow); visceral MNs in the lateral horn (box 2) are double positive for *Fbn2* and *Chat*, and skeletal MNs in the ventral horn (box 1) are only *Chat*+. **f** ISH for novel marker gene *Tns1* (cyan) highlights skeletal MNs, showing restricted localization to the ventral horn and co-expression with Chat in cells with the distinctive cell body shape of skeletal MNs. Co-labeling with *Prph* (red) and *Chat* (yellow) highlights skeletal MNs in the ventral horn (box 1) that are triple positive, visceral MNs in the lateral horn (box 2) that are double positive for *Chat* and *Prph*, and interneurons in the intermediate zone (box 3) that are only positive for *Chat* (arrow) or occasionally double-positive for *Chat* and *Prph* (arrowhead). **e**, **f**: thoracic spinal cord; all images are merged for the 3 colors except for *Fbn2* and *Tns1*, shown alone, as marked. Full single-color images and merges are shown in Supplementary Fig. 6. Low magnification image scale bars, 200 μm. High magnification scale bars, 50 μm.

motor neurons. Therefore, although they were not *Pax2*-positive, we suspect I8 represents a rare type of cholinergic interneuron. Meanwhile, *Zeb2*, a marker for visceral motor neurons[23], was most highly expressed in neighboring clusters we named V1–V10. Notably, one cluster of presumptive visceral MNs based on its location in the UMAP, V8, had undetectable *Zeb2* expression, but

instead shared expression of the gene *Fbn2* with the other visceral clusters, identifying this gene as a more general candidate marker for visceral MNs (Fig. 2c–d; Supplementary Fig. 5c). To test this prediction, we examined the expression of this gene in spinal cord sections using multiplexed fluorescent in situ hybridization (ISH; Fig. 2e; Supplementary Fig. 6c) relative to all cholinergic neurons

(*Chat*+) and most inhibitory neurons (*Slc6a1*+). Notably, *Fbn2* expression localized to a small group of *Chat*+/*Slc6a1*− neurons in the lateral column of the spinal cord where visceral MNs reside, confirming the value of *Fbn2* in identifying the majority of visceral motor neurons (Fig. 2d, e; Supplementary Fig. 12).

By default, the remaining three clusters lacking *Pax2* or *Zeb2* must include skeletal MNs, for which selective markers were not known. These clusters account for a significant fraction of cholinergic neurons (Supplementary Fig. 5b). Notably, the gene *Tns1* was strongly expressed in all three clusters and was essentially absent from other types of cholinergic neurons (Fig. 2b–d). ISH demonstrated that *Tns1* is robustly and selectively expressed in a group of large neurons in the ventral horn of the spinal cord, exactly where the cell bodies of skeletal MNs are located (Fig. 2f; Supplementary Fig. 6b, d). The *Tns1*-positive neurons invariably co-expressed two well-characterized markers of cholinergic neurons, *Chat* and *Prph*[24,25] (Fig. 2f; Supplementary Fig. 6a, b), confirming that *Tns1*-positive cells are indeed skeletal MNs. Moreover, all *Chat*-positive neurons in this area of the ventral horn co-expressed *Tns1*, thereby defining *Tns1* as a selective marker for skeletal MNs. In combination, these data also demonstrate that the three clusters S1–S3 represent transcriptomically distinct classes of this important type of motor neuron.

**Skeletal motor neurons.** Skeletal muscle is composed of extrafusal fibers, which generate force for locomotion, and intrafusal fibers, which contain muscle spindles that detect muscle stretch and facilitate contraction. Three subtypes of skeletal MNs innervate skeletal muscle including well-characterized alpha and gamma subtypes with distinct electrophysiological properties, and a third less well-defined class, beta[26]. Alpha motor neurons have a large cell body diameter and exclusively innervate extrafusal muscle fibers and drive muscle contraction. Gamma motor neurons innervate intrafusal fibers, regulating muscle spindle sensitivity to stretch. Beta motor neurons are thought to innervate intrafusal, as well as extrafusal muscle fibers and to share properties with both alpha and gamma MNs[27,28]. Although the size, presence, or absence of C-boutons and expression of markers (see below) are often used as proxies to assign MN subtypes, electrophysiology is required for definitive characterization. Ample evidence shows that motor neuron disease more severely affects some subtypes of skeletal MNs[5,7]. For example, in ALS, fast-firing alpha MNs that innervate fast fatigable muscle fibers are widely described to degenerate first, both in patients and in animal models[8,29,30], while slow-firing alpha MNs and gamma MNs are more resistant[7,31]. Discovering specific markers for each skeletal MN type and subtype would shed light on this differential vulnerability, for example by facilitating their detection and allowing their manipulation in disease models.

We hypothesized that each of the three transcriptomic clusters S1–S3 (Fig. 2a, b, d) corresponds with one of the three functional types of skeletal motor neurons: alpha, beta, and gamma. Previous studies have defined *Rbfox3* as a selective marker for alpha MNs, and *Esrrg* and *Gfra1* for gamma MNs[32–34]. There are no known markers for beta MNs. Of the three clusters (S1, S2, S3), only cluster S2 strongly expressed *Rbfox3*, tentatively assigning to it the alpha MN identity (Fig. 3a). Clusters S1 and S3 both expressed gamma markers *Esrrg* and *Gfra1*, with S3 showing slightly higher *Gfra1* expression (Fig. 3a), indicating S3 might represent gamma MNs. S1 shared transcriptional similarities with both S2 (Fig. 3b) and S3 (Fig. 3a; Supplementary Fig. 7c–e). Several other genes reported as markers of skeletal subtypes[35–37] did not appear restricted to any one cluster by

snRNAseq (Supplementary Fig. 7c–e), perhaps reflecting the sensitivity of the single nuclear sequencing approach.

The expression profiles of previously identified markers support the assignment of S2 and S3 as alpha and gamma motor neurons, respectively, but suggest that, of these markers, only *Rbfox3* is likely to be sub-type specific. The third type, S1, expresses markers of both alpha and gamma, perhaps consistent with it representing the less well-characterized beta MN. However, given the lack of definitive markers for these neurons and the need to demonstrate their properties physiologically and anatomically[38], these will be referred to as Type 3 MNs. Since snRNAseq provides a rich resource for studying gene-expression profiles of different cell types, we next used our data to identify a series of strong candidate markers for the three classes of skeletal motor neurons (Fig. 3b). For example, we predicted *Sv2b, Stk32a*, and *Glis3* would be exclusively expressed in alpha MNs, *Nrp2* in gammas, and *Gpr149* in Type 3 (Fig. 3b, c; Supplementary Fig. 8e, f). Moreover, *Rreb1* would be primarily expressed in Type 3 MNs with weaker expression in alphas, and *Plekhg1* primarily expressed in gammas with lower expression in Type 3 (Supplementary Fig. 8f).

Multiplexed ISH revealed candidate alpha markers *Sv2b* and *Stk32a* are co-expressed with known marker *Rbfox3* in cholinergic neurons in the ventral horn (Fig. 3d). Notably, these cells also exhibit large diameters typical of alpha MNs, anatomically diagnostic C-boutons apposed to their cell bodies (Fig. 3e), as well as VGLUT1-positive synapses (Supplementary Fig. 9a). These molecular and anatomic features strongly validate our assignment of this class of cells as alpha MNs. By contrast, a population of smaller diameter skeletal motor neurons expressed the candidate Type 3 marker *Gpr149* mRNA and lacked prominent C-boutons (Fig. 3e; Supplementary Fig. 8g). Similarly, we identified a different group of smaller diameter skeletal MNs expressing the gamma marker *Nrp2*, with a subset of these co-expressing *Plekhg1*, albeit at lower levels (Supplementary Fig. 8c, d), and these neurons lacked C-boutons (Fig. 3e). Importantly, our strategy for snRNAseq of cholinergic neurons uncovered many genes for distinguishing between skeletal MNs subtypes. The three types may exhibit different susceptibility in disease, but their specific detection has been hampered by the small number of existing markers. Our characterization of distinguishing markers (Fig. 3d–e; Supplementary Fig. 8a–e) provides an approach to identify all three skeletal MN types. Future localization of other genes with specific expression patterns (Fig. 3b; Supplementary Data 1) may suggest complementary strategies and uncover even better diagnostic probes. Moreover, *Sv2b* and *Stk32a* are highly specific for alpha MNs and thus represent a significant advance over *Rbfox3* since their expression pattern is much more restricted to motor neurons in the spinal cord (Supplementary Fig. 14u, z, a'). Finally, we were able to validate the alpha maker *Sv2b*, encoding synaptic vesicle glycoprotein 2B, at the protein level, by SV2B immunostaining in the spinal cord and muscle, highlighting alpha motor neuron cell bodies and axon terminals, respectively (Supplementary Fig. 9b).

**Subtypes of alpha motor neurons.** Clustering of cholinergic snRNAseq data provides strong evidence that the most prominent transcriptomic differences between skeletal MNs account for their division into 3 groups. In disease, gamma MNs are thought to be more resilient than alpha MNs, and certain alpha motor neurons die whereas others survive[9,31,39–41]. Because of this differential vulnerability, we examined what types of distinctions were reflected in the transcriptome by re-clustering the snRNAseq data from just the skeletal MNs (Supplementary Fig. 7a, b)

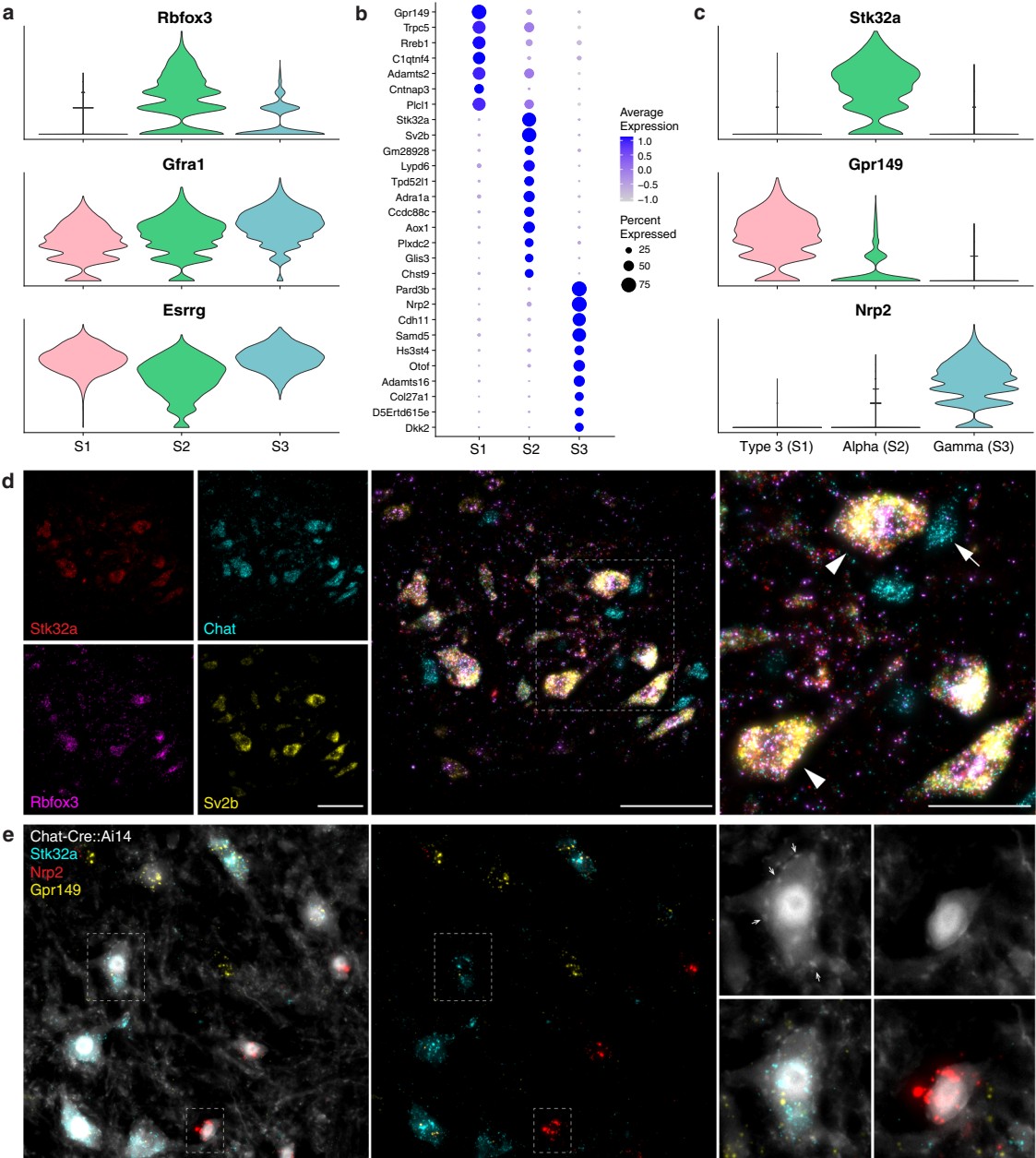

**Fig. 3 Novel markers for distinguishing alpha, gamma, and Type 3 motor neurons. a** Expression of known alpha and gamma markers by skeletal clusters. **b** Dot plot for the top marker genes for each skeletal cluster, showing more shared expression between clusters S1 and S2. **c** Expression of novel and specific marker genes for each skeletal MN subtype. **d** ISH showing expression of novel alpha MN markers *Stk32a* (red) and *Sv2b* (yellow) co-expressed with known alpha marker *Rbfox3* (magenta) in large diameter *Chat+* (cyan) ventral horn neurons. Arrowheads highlight examples of large diameter, triple-positive alpha MNs; arrow highlights the example of a small diameter *Chat+* MN that is negative for all 3 alpha markers (i.e., gamma or Type 3). **e** ISH image overlayed on endogenous Chat-Cre::Ai14 labeling showing expression of S1 marker *Gpr149* (yellow), S2 marker *Stk32a* (cyan), and S3 marker *Nrp2* (red). High magnification images highlight a large MN expressing *Stk32a* and surrounded by C-boutons (examples indicated by arrows), characteristic of alpha MNs, and a small MN expressing *Nrp2* and lacking C-boutons, characteristic of gamma MNs. ISH from the ventral horn of lumbar spinal cord sections. Low magnification scale bars, 100 µm. High magnification scale bars, **d** 50 µm, **e** 20 µm.

and just the alpha MNs. We observed 8 subtypes of alpha MNs (Fig. 4a) that we hypothesized might correspond to structural or functional features, such as innervation patterns or electrical properties. Because our experimental design included differential barcoding of nuclei from the cervical, thoracic, and lumbar/sacral cord to examine the distribution of cholinergic types along the length of the spinal cord (Fig. 1; Supplementary Fig. 5d), we explored the localization of alpha MNs among these regions (Fig. 4b). We noted that 3 of the alpha MN subclusters were primarily present in the cervical region (clusters 2, 6, and 7), suggesting these might correspond to very specific alpha MN types, such as the neurons innervating the diaphragm. We also noted that two types (clusters 3, 4) were present only in cervical and lumbar regions and were absent from the thoracic segment, suggesting they might correspond to alpha MNs that innervate specific parts of the limbs. Finally, there were multiple subtypes (clusters 0, 1, 5) that were closely related to one another and were present at all three levels.

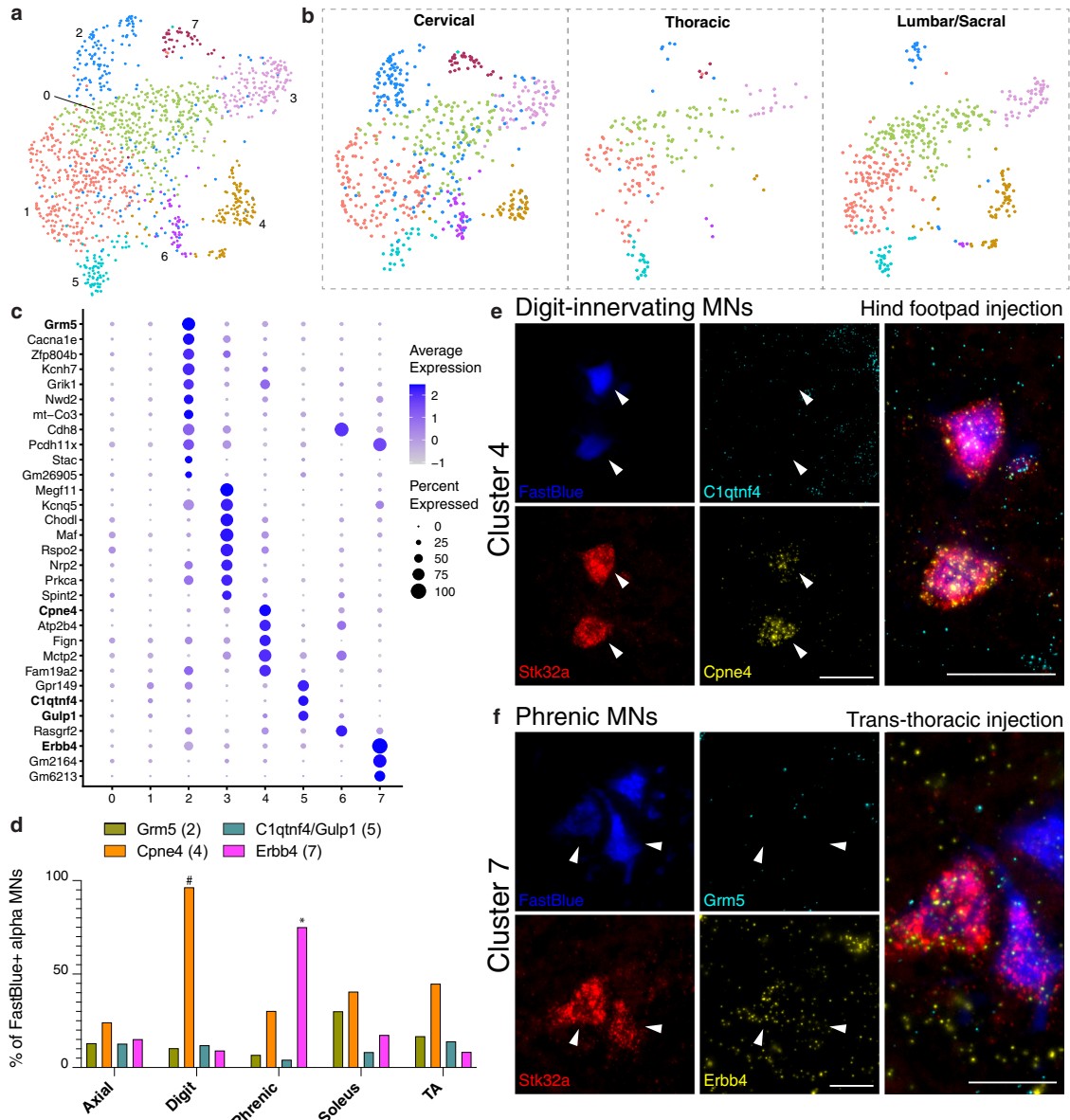

**Fig. 4 Alpha motor neurons sub-cluster into eight transcriptional groups. a** Alpha MNs can be re-clustered into 8 sub-clusters. **b** UMAPs displaying the alpha MN clusters in each separately barcoded segment of the spinal cord. Clusters 2 and 7 are most prominent in cervical, 3 and 4 are sparse or missing from thoracic. **c** Dot plot showing the top marker genes for each cluster. No markers were identified for clusters 0 or 1. Genes in bold are shown by ISH. **d** Quantification of cluster marker expression in alpha MNs retrogradely labeled from various muscles (Axial = lumbar extensors of the spine, TA = tibialis anterior). For each muscle, N = 2 animals were injected with the retrograde tracer FastBlue. Percent of FastBlue+ alpha MNs, based on *Stk32a* expression, expressing cluster markers from each injected muscle. *Cpne4* (axial n = 66 neurons, digit n = 56, phrenic n = 175, soleus n = 91, TA n = 78) and *Erbb4* (axial n = 138 neurons, digit n = 77, phrenic n = 177, soleus n = 63, TA n = 119) stand out as markers for digit-innervating and phrenic MNs, respectively. *Grm5* was relatively enriched in alpha MNs projecting to the soleus compared with the other muscles (axial n = 138 neurons, digit n = 77, phrenic n = 177, soleus n = 63, TA n = 119), and *C1qtnf4/Gulp1* was expressed in double the number of alpha MNs projecting to digit muscle (axial n = 94 neurons, digit n = 67, phrenic n = 142, soleus n = 72, TA n = 100). **e** ISH examples of images quantified in d. Alpha MNs innervating the digits express the cluster 4 marker *Cpne4* (yellow) but lack the cluster 5 marker *C1qtnf4* (cyan). **f** Alpha MNs innervating the diaphragm (phrenic MNs) express the cluster 7 marker *Erbb4* (yellow) but lack the cluster 2 marker *Grm5* (cyan). *p = 0.0053 by Shapiro-Wilk test comparing by muscle. #p = 0.0031 by Shapiro-Wilk test comparing by the gene. Scale bars, **e** 50 μm, **f** 25 μm.

We determined a panel of diagnostic markers for 4 of the 8 subtypes of alpha MNs (bold font in Fig. 4c, d) to examine their expression in alpha MNs innervating several different muscles. We injected the muscles with a retrograde tracer, fixed and imaged the spinal cords to identify retrogradely labeled cells, then performed in situ hybridization to detect the mRNA markers in labeled *Stk32a*-positive alpha motor neurons (Fig. 4 e, f; Supplementary Fig. 10). We found that *Cpne4* expression

(cluster 4) correlated well with digit muscle-innervating neurons (Fig. 4d, e), as previously reported[42]. Further, we discovered that *Erbb4*-expressing alpha MNs project via the phrenic nerve to the diaphragm (cluster 7; Fig. 4d, f). However, among the remaining probes we tested, we did not detect any correlation with the muscle types injected, including the lumbar extensors of the spine[43] (axial), soleus, and tibialis anterior muscles.

We performed an analysis to examine the types of genes enriched in the subtypes that could impact firing properties and found several candidates for each subtype (Supplementary Data 2). Future electrophysiological studies, coupled with our dataset, will provide a more thorough understanding of subtype-specific firing properties.

In combination, our analysis of gene expression in skeletal MNs defined markers for the three main classes, revealed that each of these can be divided into subtypes, and identified transcriptomic differences between alpha MNs that are likely to correspond to innervation of distinct muscles with specific functional properties. In particular, we identify a combination of markers for the subtype of alpha MN that innervates the diaphragm, the loss of which ultimately causes mortality in ALS.

**Cholinergic interneurons**. Cholinergic interneurons are found in the intermediate zone of the spinal cord where they modulate circuit function to coordinate locomotor behavior[3], but they have not been extensively studied. The best characterized cholinergic interneurons, also called partition cells, reside near the central canal and express *Pitx2*. Elegant studies have demonstrated they are the source of cholinergic C boutons that synapse either ipsi- or contralaterally onto motor neuron cell bodies and modulate their excitability[2,3,44–46]. In our dataset, we found 8 interneuron clusters with different transcriptomic profiles and specific markers (Fig. 5a). We subclustered these neurons and distinguished 14 subclasses that did not exhibit differences along the rostrocaudal axis (Supplementary Fig. 11). Although a few potential markers were identified, many were shared between the new divisions, therefore we carried out our analysis on the simpler clustering of cholinergic interneurons (Fig. 5a). Two of these classes, I5 and I6, express *Pitx2* (Fig. 5g), and are distinguished by the expression of *Tox* in I5 but not I6 (Fig. 5h). As predicted by the snRNAseq data, we identified examples of these two types of partition cells by ISH (Fig. 5c).

Although much more is known about partition cells, the majority of cholinergic interneurons did not express *Pitx2* (Fig. 5g). Four clusters, I1, I2, I3, and I4, accounted for almost all other cholinergic interneurons, indicating that these interneurons are much more diverse than was previously known. These cells were selectively labeled by *Slc6a1*, a GABA transporter (Fig. 2c, d), and *Mpped2* (Fig. 5b; Supplementary Fig. 5c). I8 is a small group of cholinergic neurons sharing an expression of *Mpped2* with I1-I7 (Figs. 2c, 5b; Supplementary Fig. 5c). To validate that this cluster represents a rare type of interneuron, we defined a specific combination of markers to localize I8 (Fig. 5d). As predicted from the sequencing, we demonstrated the existence of cholinergic neurons in the intermediate zone that co-expressed. Diagnostic markers for I8, *Piezo2* and *Reln*, in a subset of *Chat+* neurons (Fig. 5e, f). These cells localized to the intermediate zone around the central canal (Fig. 5d), substantiating their role as an unrecognized type of cholinergic interneuron.

The expression of *Piezo2* in cholinergic interneurons was surprising as this mechano-sensitive ion channel is primarily expressed in peripheral sensory neurons, where it is required for sensitivity to touch stimuli to the skin[47,48], aspects of interoception (e.g., breathing[49]), and proprioception[48]. We also detected its expression in other interneuron clusters (I1, I7; Fig. 5e), a particular subtype of alpha MN (subcluster 6), as well as several visceral MN-clusters (V2, V3, V7, V8; Supplementary Fig. 14r). It will be of great interest to examine Piezo2 function in these different neuron types and to determine the types of internal mechanical stimuli that it may allow these cells to detect.

**Visceral motor neurons**. The pre-ganglionic autonomic MNs, or visceral MNs, is the third main class of cholinergic cells. These neurons provide motor control in the autonomic nervous system and are responsible for relaying CNS messages that control not only involuntary movement of smooth muscles and glands but also many immediate physiological responses including heart rate, respiration, and digestion[50]. Unlike other spinal motor neurons, visceral MNs do not directly innervate their final effector organ, acting instead via ganglia in the periphery. Reclustering visceral MNs in our dataset divided them into 16 subclusters (Fig. 6a) that were distinguished by select markers (Fig. 6b). Intriguingly, in addition to transcription factors, many of the markers we identified that best describe each visceral MN cluster include neuropeptides (*Ccbe1*, *Sst*, *Penk*) and genes involved with their production (*Pcsk2*). Neuropeptides modulate the function of many cells and processes, thus their differential expression in subsets of visceral MNs likely reflects the diverse functions and organs that these neurons control. Other distinguishing markers of visceral MN subtypes included secreted molecules (*Fam19a1* and *Fam163a*) and extracellular matrix proteins (*Postn*, *Fras1*, *Reln*, *Mamdc2*) further highlighting how their diverse gene-expression profiles are likely a reflection of their distinct physiological roles.

Visceral MNs are restricted to the pre-ganglionic column in the lateral horn and thought to be primarily found at thoracic and sacral levels[1]. Moreover, immunolocalization of neuropeptides suggested that different subtypes might exist and show restricted localization along the length of the spinal cord[51]. Remarkably, our sequencing revealed that visceral motor neurons are abundant not only in the thoracic and lumbar/sacral spinal cord but also in the most caudal cervical region (Fig. 6c; Supplementary Fig. 13; Supplemental Movie 2). The cervical spinal cord is canonically thought to be devoid of this type of neuron[50], though a few studies have identified visceral MNs in caudal cervical regions[52,53]. Using ISH we confirmed that diagnostic markers for visceral MNs, *Fbn2*, *Chat*, and *Prph*, are co-expressed in the lateral column of cervical spinal cord sections (Fig. 6f; Supplementary Fig. 12a, 13e–h). Therefore, this important class of cholinergic neurons appears more widely distributed than previously suspected.

Moreover, the distribution of visceral MN clusters was strikingly diverse between the cervical, thoracic, and lumbar/sacral spinal cord (Supplementary Fig. 5d; Fig. 6c). Select clusters were entirely missing from certain spinal cord regions, or only present in a single region, strongly suggesting functional specialization of the neurons controlling specific organs. For example, *Dach2*, *Gpc3*, and *Sema5a* are top markers enriched in C and T levels but not lumbo-sacral, whereas *Sst* is enriched in the L/S segment but absent from C and T levels (Fig. 6d). We quantified these markers in *Fbn2+* visceral motor neurons of the lateral horn at all levels and demonstrate that the sequencing data are predictive of the relative prevalence of each subtype per level (Fig. 6e). As another example, cluster 8 marked by expression of *Bnc2* (Fig. 6b) was only detected in cervical-derived nuclear sequence data. Importantly, ISH revealed that *Bnc2* localized to a small subset of the putative visceral MNs in the cervical lateral column (Supplementary Fig. 12a), further supporting our conclusion that this region of the spinal cord has a clear and specialized role in autonomic signaling.

At a more general level, our data strongly support a more complex role for visceral MNs in autonomic signaling than previously appreciated. Indeed, of 16 transcriptomically divergent clusters (Fig. 6b), 12 were unique to or more abundant in a given region. Thus, their anatomic and transcriptomic specialization coincide, strongly supporting a functional role for their distinct gene expression profiles. Cluster 5 represented a large and

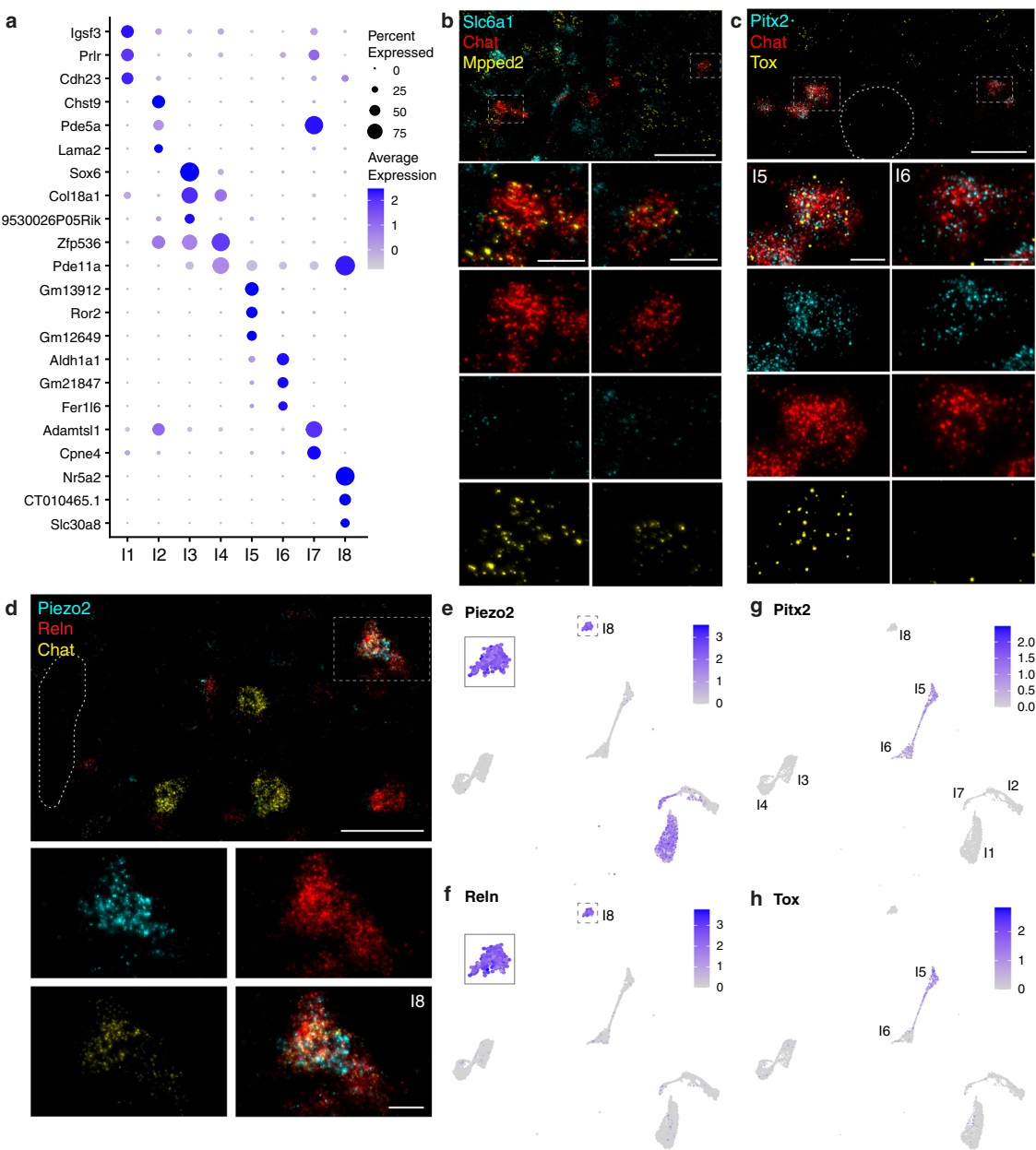

**Fig. 5 Spinal cholinergic interneurons separate into 8 transcriptionally distinct subtypes. a** Top markers of the 8 cholinergic interneuron clusters.
**b** ISH showing expression of two top markers for cholinergic interneurons (Fig. 2c), *Slc6a1* (cyan) and *Mpped2* (yellow) in *Chat*+ neurons (red) in the intermediate zone. Boxes highlight a neuron expressing *Chat* and *Mpped2*, but lacking *Slc6a1* (left), and a neuron expressing *Chat*, *Mpped2*, and *Slc6a1* (right). **c** ISH showing expression of *Pitx2* and *Tox* near the central canal (outlined) of the lumbar spinal cord. Co-expression of *Pitx2* and *Tox* in *Chat*+ cells exemplifies cluster I5 (left) whereas *Pitx2* and *Chat* without *Tox* identify a cluster I6 cell (right). **d** High-magnification images showing co-expression of *Piezo2*, *Reln*, and *Chat* in an I8 neuron found near the central canal (outlined) of the thoracic spinal cord. **e**, **f** *Piezo2* and *Reln* co-expression are diagnostic of a rare population of interneurons, cluster I8. **g** *Pitx2*, a known marker for partition cells, is expressed only in clusters I5 and I6. **h** *Tox* is expressed only in interneuron cluster I5 and thus differentiates between the two *Pitx2*-expressing clusters. Low magnification scale bars, 50 μm. High magnification scale bars, 10 μm.

particularly divergent group of visceral MNs that were present only in the lumbar/sacral region (Fig. 6c) and co-expressed the adrenergic receptor *Adra1a* with the neuropeptides proenkephalin (*Penk*) and somatostatin (*Sst*; Fig. 6b; Supplementary Data 1). Notably, ISH revealed that a large subset of visceral MNs in the sacral spinal cord express the inhibitory neuropeptide *Sst* (Fig. 6g; Supplementary Data 1), corresponding to pre-ganglionic neurons that have been previously described as controlling bladder and bowel function[54–56]. *Sst*+ *Fbn2*+ neurons of the lateral horn were absent from cervical and thoracic levels (Supplementary Fig. 12b).

## Discussion

Mammalian skeletal MNs are essential for coordinating muscle activity and all types of consciously controlled movement. They are also the cellular targets responsible for the progressive and ultimately fatal symptoms of diseases like spinal muscular atrophy and ALS. Over the past 30 years, it has become increasingly clear that diverse subtypes of skeletal MNs play distinct roles in motor control and have different susceptibility in disease. In particular, elegant developmental studies hint at considerable diversity of adult motor neurons[57] but little is known about the

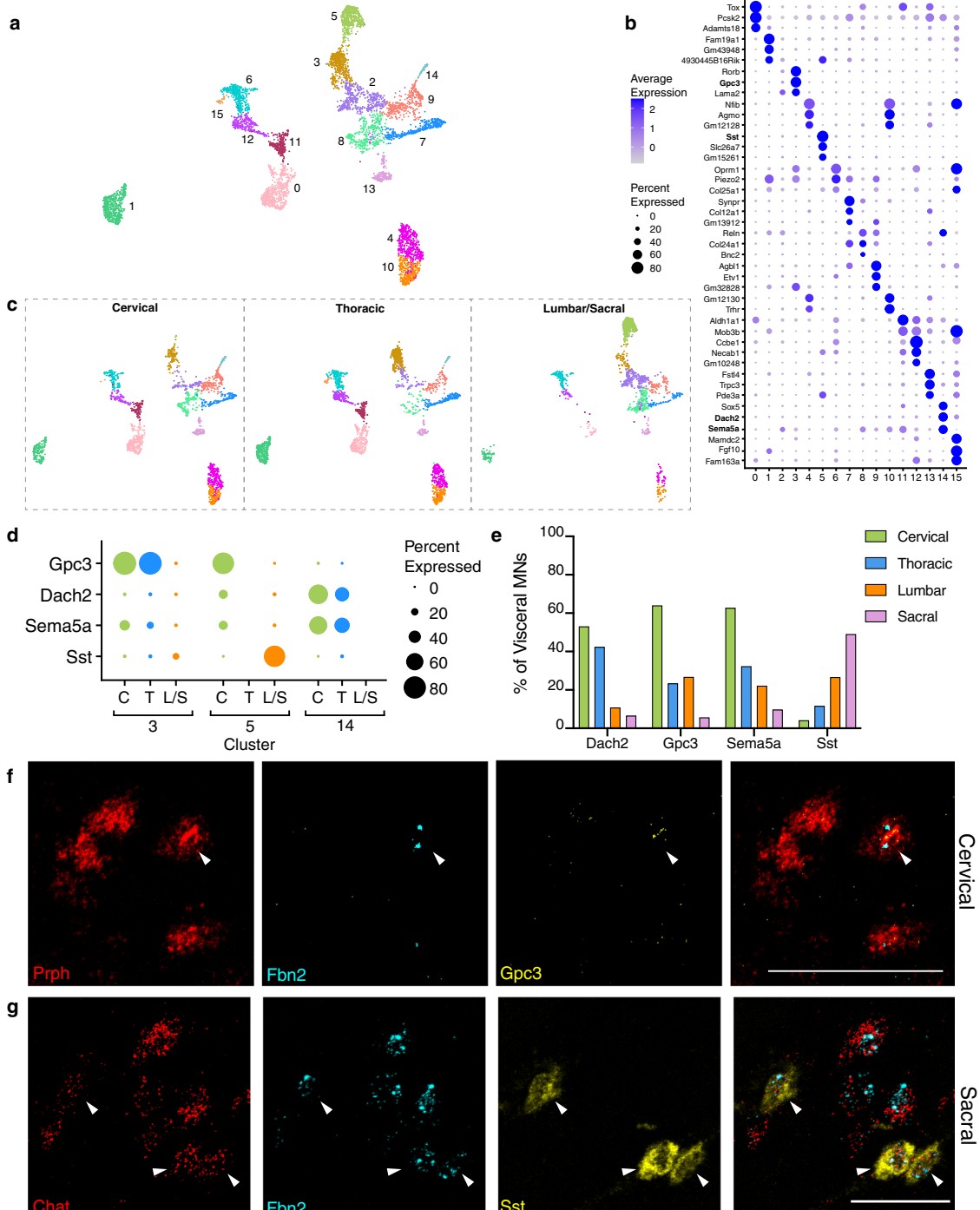

**Fig. 6 Visceral motor neurons exhibit extensive diversity in their gene expression profiles and anatomic localization. a** Visceral motor neurons can be re-clustered into 16 distinct sub-clusters. **b** Dot plot showing the top three marker genes for each cluster. Genes in bold font are shown by ISH below. **c** UMAPs displaying the visceral motor neuron clusters derived from each separately barcoded segment of the spinal cord. Clusters 0, 1, 3, 4, 10, 11, 12, 14, 15 are sparse or missing from the lumbar/sacral segment; conversely, cluster 5 is unique to the lumbar/sacral segment, where cluster 2 is also most abundant. Cluster 8 is most sparse in thoracic, while clusters 3 and 15 are most abundant in thoracic, and cluster 14 is most abundant in cervical. **d** Dot plot showing expression of marker genes for uniquely cervical/thoracic (C/T) or lumbar/sacral (L/S) clusters in each segment of the spinal cord, with *Gpc3*, *Dach2*, and *Sema5a* most highly expressed in rostral levels and *Sst* most highly expressed in caudal levels. **e** Quantification of each level-specific cluster marker in visceral MNs of each spinal cord level, shown as percent of *Fbn2* + MNs expressing each gene ($N = 2$ animals), highlighting rostral expression of *Gpc3* (cervical $n = 100$ neurons, thoracic $n = 64$, lumbar $n = 41$, sacral $n = 140$), *Dach2* (cervical $n = 175$ neurons, thoracic $n = 139$, lumbar $n = 92$, sacral $n = 91$), and *Sema5a* (cervical $n = 180$ neurons, thoracic $n = 142$, lumbar $n = 72$, sacral $n = 92$) and caudal expression of *Sst* (cervical $n = 144$ neurons, thoracic $n = 77$, lumbar $n = 75$, sacral $n = 110$), as predicted by the sequencing data. **f** ISH showing *Gpc3* + (yellow) visceral MNs, likely representing cluster 3 neurons, in the cervical spinal cord. **g** ISH showing *Sst*+ visceral MNs (yellow), likely representing cluster 5 neurons, in the sacral spinal cord. Scale bars, 50 μm.

true range of these cell types or the molecules defining them. Thus, to gain genetic access to skeletal MNs in animal models, the field has relied extensively on a Chat-IRES-Cre mouse that broadly targets cholinergic neurons. Here we used this line to mark cholinergic neurons and adopted a snRNAseq approach to characterize motor neuron diversity, identify sub-types and define a wide range of genes that can be used to selectively distinguish the many different classes of adult cholinergic neurons that we discovered. Importantly, analyzing just a subset of the sequenced nuclei does not dramatically change the clustering pattern nor the major conclusions, therefore additional sequencing has a diminishing return. Nuclear sequencing is highly representative relative to cell-based approaches but can overrepresent the expression of nuclear-retained genes and miss good cytoplasmic markers[14]. Although this is unlikely to affect clustering because this relies on many transcripts, it may mean that better markers exist.

One discovery is the previously unappreciated number of highly distinct visceral motor neuron types. That there should exist different subtypes may not be surprising, given the diversity of organs and glands they control, ranging from the heart and lungs to the adrenal medulla, intestines, and bladder. However, it was particularly compelling to find that the different types are discretely located along the length of the cord, corresponding to a body map. An especially surprising finding was that visceral MNs extends into the cervical spinal cord, where we have clearly localized at least three subtypes (Gpc3+, Dach2+, Sema5a+, and Bnc2+ cells in the lower cervical region). All previous descriptions restrict these neurons to the thoracic and sacral regions[1,50], thus it will be of great interest to further characterize these cervical visceral MNs in terms of their connectivity and function.

We also characterized a large population of cholinergic interneurons and demonstrated unanticipated diversity. Previously, only one major class of cholinergic interneurons, the Pitx2+ partition cells, has been studied at a functional level[2,3,45]. Our analysis divides the partition cells into at least two transcriptomic forms and identifies several other types of cholinergic interneurons with unknown function. Thus, the snRNAseq analysis of cells targeted by Chat-IRES-Cre mediated recombination provides a rich resource and exposes features of several unusual types of cholinergic neurons. Importantly, our data align well with a concurrent study[58] and in combination, these data should provide the field with approaches for selectively targeting cholinergic neuron types.

As we had hoped, our sequencing also identified skeletal MNs as three related but transcriptomically distinct groups. This clear division fits well with previous reports of select differences between alpha and gamma MNs and clearly demonstrates that a third type, possibly beta MNs (referred to as Type 3), which have been less well characterized, form an equally abundant class. Notably, our analysis supports these Type 3 MNs as being related to both alphas and gammas[1] but as sharing the greatest similarity with a subtype of alpha MNs. The transcriptomic description of these different MN types dramatically changes the landscape of markers that can be used to distinguish them and therefore should greatly simplify their identification. Indeed, intersectional approaches using genes we validated here and Chat, would be useful for separately targeting MN subtypes and facilitate in vivo dissection of their biological roles. Our data show that the embryonic marker code responsible for the development of distinct MN pools appears to be largely erased in the adult. Finally, each class of skeletal MN can be subdivided into several transcriptomically related but distinguishable types. For the alpha motor neurons, these divisions appear at least in part related to their peripheral targets, for example, Stk32a+ Erbb4+ alpha MNs project to the diaphragm whereas Stk32a+ Cpne4+ alpha MNs innervate digits. We anticipate that this rich transcriptomic

dataset should help define and characterize the different subtypes of MNs and expose their differential significance in health and disease.

## Methods

**Animals**. Animal care and experimental procedures were performed in accordance with protocols 17-003 and 20-003 approved by the *Eunice Kennedy Shriver* National Institute of Child Health and Human Development Animal Care and Use Committee. CAG-Sun1/sfGFP mice (B6;129-Gt(ROSA)26Sor^tm5(CAG-Sun1/sfGFP)Nat/J; Stock No: 021039[15]) were bred to the Chat-IRES-Cre::deltaNeo line (Chat^tm1(cre)Lowl/J); Jax Stock No. 031661[59]), in which the neomycin cassette was removed to avoid ectopic expression sometimes observed in the ChAT-IRES-Cre line. This cross resulted in the expression of the SUN1 fusion protein (nuclear-membrane targeting sequences of the SUN1 protein fused to 2 copies of superfolder GFP (sfGFP) followed by 6 copies of Myc) in cells that express Chat.

**Single nucleus isolation and sequencing**. Spinal cords were rapidly extruded from Chat-IRES-Cre::CAG-Sun1/sfGFP mice after anesthesia with 2.5% tribromoethanol (0.5 ml/25 g body weight), decapitation, and a cut through the spinal column at hip level. A PBS-filled syringe fitted with a 20 G needle (7.25 mm) was inserted into the caudal end of the spinal column to flush out the spinal cord. Each cord was separated into cervical, thoracic, and lumbar/sacral regions, and matching regions from 2 mice were pooled for homogenization and nuclear isolation. FACS sorted GFP+ nuclei from n = 12 mice (6 females and 6 males, 8 weeks old) were pooled from each region to obtain the nuclei analyzed in this study. Samples were frozen until processing, which was performed in 3 replicates—one of male tissue, one female, and one of mixed-sex.

The nuclei isolation protocol was adapted from Sathyamurthy et al.[12]. Each sample was thawed and homogenized in a Dounce Homogenizer (Kimble Chase 2 ml Tissue Grinder) containing 1 ml freshly prepared ice-cold lysis buffer (low sucrose buffer [320 mM sucrose, 10 mM HEPES-pH 8.0, 5 mM $CaCl_2$, 3 mM Mg-acetate, 0.1 mM EDTA, 1 mM DTT] with 0.1% NP-40) applying 10 strokes with the A pestle followed by 10 strokes with the B pestle. The homogenate was filtered through a 40 µm cell strainer (FisherScientific #08-771-1), transferred to a DNA low bind 2 mL microfuge tube (Eppendorf, #022431048), and centrifuged at 300×g for 5 min at 4 °C. The supernatant was removed, the pellet was gently resuspended in low sucrose buffer and centrifuged for another 5 min. The nuclei were resuspended in 500 µl 1× PBS with 1% BSA and 0.2 U/µl SUPERaseIn RNase Inhibitor (ThermoFisher, #AM2696) and loaded on top of 900 µl 1.8 M Sucrose Cushion Solution (Sigma, NUC-201). The sucrose gradient was centrifuged at 13,000×g for 45 min at 4 °C. The supernatant was discarded, the nuclei were resuspended in 500 µl Pre-FACS buffer (1× PBS with 1% BSA, 0.2 U/µl SUPERaseIn RNase Inhibitor), and filtered through a 35 µm cell strainer (Falcon #352235). Before FACS sorting, 2.5 µl of 5 mM DRAQ5 (ThermoFisher #62251) were added.

Samples were processed on a Sony SH800 Cell Sorter with a 100 mm sorting chip and GFP+/DRAQ5+ nuclei were collected into 1.5 ml centrifuge tubes containing 10 µl of the Pre-FACS buffer. We collected ~24,000, 15,000, and 34,000, GFP+ nuclei from the cervical, thoracic, and lumbar/sacral spinal cord samples, respectively (these nuclei were pooled from n = 4 mice in 3 replicates, n = 12 total). Using a Chromium Single Cell 3′ Library and Gel Bead Kit v3 (10X Genomics), GFP + nuclei were immediately loaded onto a Chromium Single Cell Processor (10X Genomics) for barcoding of RNA from single nuclei. Sequencing libraries were constructed according to the manufacturer's instructions and resulting cDNA samples were run on an Agilent Bioanalyzer using the High Sensitivity DNA Chip as quality control and to determine cDNA concentrations. The samples were combined and run on an Illumina HiSeq2500 with Read1 = 98-bp, Read2 = 26-bp, and indexRead = 8. There were a total of 410 million reads passing the filter. Reads were aligned and assigned to Ensembl GRm38 transcript definitions using the CellRanger v3.1.0 pipeline (10X Genomics). The transcript reference was prepared as a pre-mRNA reference as described in the Cell Ranger documentation.

**Single nucleus analysis**. Sequencing data were analyzed using the R package Seurat version 3.1.4[60] following standard procedures[61]. Outliers were identified based on the number of expressed genes and mitochondrial proportions and removed from the data. Removal of outliers resulted in 34,231 total remaining cells for analysis. Each separately barcoded region of the spinal cord was separately processed by the standard methods. Briefly, the data were normalized and scaled with the SCTransform function, the linear dimensional reduction was performed on scaled data, and significant principal components (PCs) were identified using the elbow method. Only significant PCs, (10–30, determined via the elbow plot method for each analysis) were used for downstream clustering. Clustering was performed using the Seurat functions FindNeighbors and FindClusters (resolution = 0.6). Clusters were then visualized with t-SNE or UMAP[62]. Reference anchors were identified between each spinal cord region dataset before integration with the IntegrateData function, and integrated data were then processed by the same methods. For subclustering, clusters of interest for each subtype were taken as a subset from the cholinergic neuron dataset and significant PCs were used for downstream clustering similarly to above. All data was visualized with the SCT

assay, and plots were generated using Seurat functions. Clustering of all cholinergic neurons resulted in 23 clusters (Supplementary Fig. 5a); due to their proximity, clusters 3, 4, and 22 were combined.

Cluster-specific marker genes were identified using the FindAllMarkers function, utilizing a negative binomial distribution (DESeq2[63]). Only positive markers were identified, and the data were down-sampled to 100 cells per cluster to facilitate comparison. Genes had to be detected in a minimum of 25% of cells and display at least a 50% log fold change. Only markers with a p-value under 0.1 were returned. In selecting top markers for each cluster, we prioritized expression in a low number of cells outside each cluster and a higher log fold change in a large number of cells within each cluster. We filtered the markers to identify genes with expression in fewer than 30% of cells outside of each cluster and an average log fold change greater than 60% between each cluster and all other clusters and selected the genes that were least expressed outside of each cluster for visualization. In this way, we were able to select markers that are highly expressed within each cluster, while still being restricted to genes unique to each individual cluster.

**Identification of activity-related genes among alpha MNs**. Gene ontology (GO) terms related to firing properties were selected (channel activity, GO:0005216; GABA receptor activity, GO:0016917; g-protein-coupled receptor activity, GO:0004930), and alpha MN marker genes within each ontology were identified (Supplementary Table 2).

**Fixed tissue harvest and immunostaining**. Mice were anesthetized with 2.5% avertin and transcardially perfused with saline, then 4% paraformaldehyde, followed by overnight post-fixation and cryoprotection in 30% sucrose prior to sectioning. Sixteen micrometer thick coronal slices were collected onto positively charged slides using a Leica CM3050 S Research Cryostat. For immunostaining, tissue was permeabilized in 0.1% Triton-X100 in 1× PBS (PBSTx), then blocked in 5% normal donkey serum in 0.1% PBSTx. The primary antibody was diluted in 0.5% normal donkey serum in 0.1% PBSTx and tissue was incubated overnight at 4 °C. Tissue was washed in 0.1% PBSTx and incubated in secondary antibody (ThermoFisher) diluted in 0.1% PBSTx for 1 h, washed in 1× PBS, and coverslipped with Prolong Diamond (ThermoFisher #P36961). Primary antibodies: anti-SV2B (Synaptic Systems# 119 102, 1:500), anti-VGLUT1 (Synaptic Systems #135 011, 1:500). Alpha-bungarotoxin (ThermoFisher #B13422, 1:200) was diluted with a secondary antibody.

**Retrograde tracing**. Motor neurons innervating specific muscles were retrogradely labeled using the tracer FastBlue (Polysciences Inc. #17740) at 5% w/v. Mice were anesthetized using isofluorane and a small incision was performed on the skin to expose the muscle of interest. Tracer was loaded into a Hamilton syringe (Hamilton #1701N) and mounted onto a syringe pump (KD Scientific #78-0220). Tracer was delivered at a rate of 1 μl/min, 1 μl per site, 2–4 sites per muscle, depending on muscle size. Tibialis anterior (TA) was injected at 3 sites per side, soleus at 2 sites per side, lumbar extensors of the spine[43] (axial) at 4 sites per side, hindlimb footpads (digit) at 3 sites per side. Phrenic MNs were labeled by trans-thoracic injection, as previously described[64]. The tissue was fixed and harvested 3 days post-injection.

**Multiplexed in situ hybridization**. Spinal cord tissue was rapidly extruded as above or dissected out, immediately embedded in O.C.T. compound (Tissue-Tek), and fresh frozen on dry ice, taking care to work rapidly in order to minimize RNA degradation. The tissue was cut into cervical, thoracic, lumbar, and sacral regions that were co-embedded together for parallel processing and stored at −80 °C. Blocks were sectioned into 16 μm-thick coronal slices onto positively charged slides using a Leica CM3050 S Research Cryostat. Slides were dried in the cryostat, then stored at −80 °C for up to 2 weeks. Multiplexed in situ hybridization was performed according to the manufacturer's instructions for fresh frozen sections (ACD: 320851). Briefly, sections were fixed in 4% paraformaldehyde, treated with Protease IV for 30 min, and hybridized for 2 h at 40 °C (HybEZ II System) with gene-specific probes to mouse *Bnc2, C1qtnf4, Chat, Cpne4, Dach2, Erbb4, Fbn2, Glis3, Gpc3, Gpr149, Grm5, Gulp1, Mpped2, Nrp2, Piezo2, Plekhg1, Prph, Rbfox3, Reln, Rreb1, Sema5a, Slc6a1, Sst, Stk32a, Sv2b, Tns1, Tox,* and *Zeb2* identified from single nucleus analysis. Each probe was tested in at least *n* = 2 mice.

Sections for in situ hybridizations of fixed tissue were harvested and sectioned as previously described. Slides were dried at 60 °C for 10 min, then stored at −80 °C for up to 2 weeks. For in situ images overlayed with Fast Blue or Chat-Cre:: Ai14 signal, multiplexed in situ hybridization was performed according to the manufacturer's instructions for fixed frozen sections (ACD: 323100, 323120) through dehydration, then imaged in a 2× SSC buffer with pyranose oxidase (Sigma: P4234) and RNAse inhibitor (NEB: M0314L), after which the ACD protocol resumed. Probe targets were visualized using Opal dyes 520, 570, 620, or 690 (Akoya). Catalog numbers for all in situ hybridization reagents are listed in Supplementary Table 2.

Slides were imaged either using a Zeiss Axiocam 506 color camera and Zeiss Apotome.2 for optical sectioning, or a Zeiss confocal LSM800. Imaging settings were stitched using Zeiss ZEN software (blue edition). We used FIJI[65] to generate maximum intensity projections and adjust brightness and contrast.

**Overlaying tracer or endogenous signal with multiplexed ISH signal**. Images from before (FastBlue or Ai14 signal) and after (ISH signal) probe hybridization were overlaid and manually aligned with the Arivis Vision 4D software channel shift tool, using unique cell shapes, tissue tears, and section edges as landmarks. Channels were saved separately and merged using FIJI software.

**Quantification of cells labeled by ISH markers (FastBlue retrograde label experiments, and analysis of *Fbn2*+ lateral horn cells)**. Using FIJI software, FastBlue labeled cells were outlined manually. Cells were counted as positive for in situ hybridization probes if there were 3 or more distinct puncta within the region defined by FastBlue. Only FastBlue labeled cells also positive for *Stk32a* were used in the analysis. A Shapiro-Wilk test was performed on the ratio of probe-positive cells within *Stk32a*+ cells for each cluster marker by muscle. Probes were tested in at least 2 animals per muscle.

*Chat* or *Prph* signal in the lateral horn was used to determine visceral MN regions of interest. The *Chat*+ or *Prph*+ cells were outlined, and those that were also *Fbn2*+ were analyzed for expression of marker genes expected to display differential localization along the rostro-caudal axis of the spinal cord. Positivity was determined by the identification of 3 or more distinct puncta within the outlined region of each cell. The percent of visceral subcluster marker (*Dach2, Gpc3, Sema5a*, or *Sst*) positivity in these cells was determined for each spinal cord level (C, T, L, and S).

**Whole tissue immunolabeling and clearing by iDisco+**. Whole adult spinal cords were dissected out from Chat-IRES-Cre::CAG-Sun1/sfGFP double hetero-zygous mice and fixed in a straightened position using 4% paraformaldehyde in PBS. To immunolabel and visualize GFP-positive nuclei within whole cleared tissue, spinal cords were processed for iDisco+ as previously described[66]. Spinal cords were immunolabeled with Chicken anti-GFP (ThermoFisher #A10262) at 1:200 during a 3-day incubation at 37 °C, washed, then labeled with AlexaFluor 647-conjugated goat anti-chicken secondary antibody (ThermoFisher #A21449) at 1:200 for 2 days at 37 °C. Spinal cords were embedded in 1% agarose prior to the clearing steps to facilitate their handling and imaging once cleared. Imaging was performed by light-sheet microscopy using the Ultramicroscope II (Miltenyi Biotec) and the LaVision Biotec ×4 objective. Images were acquired using Imspector software (single sheet, 13 dynamic focus planes, and Contrast Adaptive/Contrast algorithm). Video and 2D image renderings were generated using Arivis Vision 4D software.

**Statistics and reproducibility**. All quantification of microscopy images was performed blinded. Statistical analyses were performed using GraphPad Prism 9. All validation of sequencing results by in situ hybridization was replicated across at least 5 sections from multiple animals. All attempts at replication were successful. All micrographs are representative images.

**Reporting summary**. Further information on research design is available in the Nature Research Reporting Summary linked to this article.

## Data availability
The datasets generated during and/or analyzed during this study have been deposited in the Gene Expression Omnibus (GEO) under accession number "GSE167597". An interactive web portal for exploring the dataset is available at www.spinalcordatlas.org and also at https://seqseek.ninds.nih.gov/. All other relevant data supporting the key findings of this study are available within the article and its Supplementary Information files or from the corresponding author upon reasonable request. A reporting summary for this article is available as a Supplementary Information file. Source data are provided with this paper.

## Code availability
Code used in analyzing nuclear sequencing data is available at https://github.com/malkasla/Alkaslasi-et-al._2021 [https://doi.org/10.5281/zenodo.4580493].

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

## Acknowledgements

We would like to thank intramural NIH colleagues Drs. Steven Coon, James Iben, and Tianwei Li of the Molecular Genomics Core (NICHD) for RNA sequencing and alignment to the pre-mRNA genome, Drs. Apratim Mitra and Ryan Dale of the Bioinformatics and Scientific Programming Core (NICHD) for technical advice, the NICHD Microscopy and Imaging Core (MIC), and Drs. Nicholas Ryba (NIDCR) and Alexander Chesler (NCCIH) for helpful discussion and comments on the manuscript. CLP is intramurally funded by the National Institutes of Health, ZIA-HD008966 (NICHD).

## Author contributions

M.R.A., Z.E.P., and C.L.P. designed the experiments and wrote the paper. M.R.A. and Z.E.P. performed computational analysis and data validation by in situ hybridization. M.R.A. performed retrograde labeling. M.R.A., Z.E.P., and S.H. performed immunostaining, imaging, and quantification of in situ hybridization experiments. T.J.P. provided advice on experimental design and expertise for sample preparation. H.S. generated the mouse crosses. L.C. and H.S. helped perform experiments. Y.Z., H.S., and T.J.P. generated the single nucleus cDNA libraries for sequencing. All authors reviewed and edited the manuscript.

## Funding

## Competing interests

The authors declare no competing interests.
