## [Peer Review File · Nature Communications]

Reviewers' Comments:

Reviewer #1:

Remarks to the Author:

This manuscript by Alkaslasi et al. describes a single nuclear RNA sequencing approach used on the adult spinal cord to identify cholinergic (Chat) classes of interneurons and motor neurons based on their transcriptional profile. The authors use a neuronal and Chat specific fluorescent sorting and to optimize their harvesting of Chat+ cells from the cord. They first identify 31 clusters that are neuronal – with 13 co-expressing either Slc176a or Gad1 – and then subtract the Slc176a or Gad1 clusters to end up with 19 (??) 'pure' Chat+ cluster. Through data mining and using previous genes in Chat+ spinal neurons as entry points, they categorize clusters as visceral preganglionic motor neurons, interneurons, gamma, beta and alpha-motor neurons (the later with fast and slow motor neuron groups). They conclude that their data provide a new and comprehensive categorization of Chat+ neurons in the spinal cord that may be used to perform targeted functional studies.

The study is very well carried out as a transcriptional analysis and because it use a single nuclear RNA sequencing approach it is likely that it has been able to capture more cells and therefore presumably more transcripts than the fair number of previous studies using single cell sequencing or bulk sequencing (especially of the large alpha-MNs that are fragile for FACS). From that point of view the study could provide important new information. My enthusiasm for the study is, however, dampened considerably because of lack of functional confirmation of any of the clusters. This leads to a number of very strong claims that are unsubstantiated (possible misleading?) and need direct confirmation.

Essentially the categorization into the three main categories (visceral motor neurons, ventral motor neurons, and interneurons) are based on unique markers identified in previous studies and then from there a subcategorization is made. This means that the categorization is biased beforehand. There are no independent means of knowing how many cells that are captured by the analysis. Such an estimate could have been obtained by retrogradely labelling motor neurons from the ventral roots combined with expression analysis. Now the only verification is the location in the cord and the soma size. But this is not enough to do the functional/anatomical verification. For example, retrograde labelling from ventral root of cells in the cervical cord is required to support the claim that the author have found a hitherto unrecognized population of visceral cells in that region. Similarly, to verify the slow and fast motor neuron markers indeed are markers or these pools of motor neurons retrograde labelling from 'fast' and 'slow' hindlimb muscles are required to directly confirm the finding which will also give information about whether motor neurons are flexor or extensors.

Also to show that a gamma motor neuron is a gamma motor neuron it is not enough to look at size – there are many indirect criteria that need to be fulfilled to say that a cell in the ventral horn is a presumptive gamma motor neuron (see Friese et al. 2009; Shneider et al. 2009) some of which the authors should used for identification (retrograde labelling, lack of Vglut1 synapses, lack of C-boutons). Admittedly this is extra work but, in my opinion, needed to make a claim about identity. To identify a gamma motor neuron directly can only be done electrophysiologically – which needs to be recognized.

The categorization as a beta motor neuron is problematic. Very few publications talk about beta-motor neurons. This is because it is extremely difficult to show a motor neuron is a beta motor neuron. It is a requirement to show that the motor neurons anatomically innervate but striatal musculature and muscle spindles (alternatively show that physiologically). The authors are not doing any test to show that and can therefore they cannot claim that they have found markers for beta-motor neurons.

In conclusion: the study is potentially very interesting showing new markers for Chat+ spinal cells.

However, the lack of independent verification of cell type categorization is a major weakness of the study which unfortunately make the strong claims about new cell markers unsubstantiated. This is unfortunate because the study could have been taken so much further. Without these experiments the study remains descriptive and suggestive with regard to cell type categorization.

Reviewer #2:

Remarks to the Author:

The work by Alkaslasi, Piccus et al. provides a molecular atlas of cholinergic neurons found in the adult mouse spinal cord. The authors used a genetic approach to permanently mark and isolate nuclei from this rare population of neurons. This by itself is an important achievement because cholinergic motor neurons are notoriously difficult to isolate from adult spinal cords. Next, they performed single-nucleus RNA-Seq on samples collected from cervical, thoracic and lumbar regions. They identified 19 distinct clusters. Based on previous markers and computational analysis, they break down these to interneurons (I1-I8), visceral MNs (V1-V8) and skeletal MNs (S1-S3). Through re-clustering, they identify substantial diversity for visceral and skeletal MNs and suggest novel markers for these cell types. Collectively, the findings provide a comprehensive transcriptomic description of spinal cholinergic neurons.

This work is of interest to the field of motor neuron biology and disease. Most conclusions are supported by evidence, unless otherwise noted (see Major Concerns). However, I am not convinced that it provides a conceptual advance to influence thinking in the field. In other words, a transcriptomic description of cholinergic neurons in the adult spinal cord is certainly valuable, but to advance the field the authors should use their molecular atlas to answer an important biological question and provide new insights, especially in light of previous reports hinting molecular diversity in spinal cord neurons. Comparing this work to others (Okaty et al., *Neuron* 2015; Sathyamurthy et al., *Cell Reports.*, 2018; Tran et al., *Neuron*, 2019), it seems that the current manuscript, after establishing a molecular atlas, does not take the next crucial step to answer a biological question. Hence, the current version of this manuscript, in my humble opinion, does not make a conceptual contribution that meets the standards of *Nature Communications*.

Major concerns

Establishing a molecular resource and then "using" it to provide new knowledge is what distinguishes a descriptive study from a substantial contribution. Since the major discovery of this paper is the identification of a dozen visceral MN subtypes, the authors could use, for example, an ALS mouse model and test whether these subtypes are differentially susceptible to disease, similar to a study on retina ganglion cells (Tran et al., *Neuron*, 2019). Evidence that some visceral MNs may be affected in ALS has been reported previously

<https://onlinelibrary.wiley.com/doi/abs/10.1002/mus.24457>

Moreover, cholinergic interneurons are also affected in a mouse model of ALS. <https://www.ncbi.nlm.nih.gov/pmc/articles/PMC3607155/>

Testing whether the identified subtypes of visceral MNs and/or cholinergic interneurons are differentially susceptible to disease would dramatically strengthen this work and its conceptual advance.

It is curious that from the 14,738 nuclei sequenced, only 6,941 were expressing ChAT. The authors suggest that the large number of excitatory and inhibitory neurons captured likely reflects that ChAT-IRES-Cre was expressed in these cells or their lineage. However, their genetic method permanently marked cholinergic cells in expected locations (lines 83-84). Fig. 1 clearly shows that the vast majority of GFP+ nuclei are indeed in these expected locations. Only a minority of GFP+ cells is outside these locations. Hence, an alternative explanation should be provided to clarify how ~8,000 of the 14,738 nuclei turned out to be excitatory and inhibitory neurons. A suggestion would be to test whether the ROSA-Sun1sfGFP mice (without Cre) show any baseline recombination events. Of note, wild-type mice, not ROSA-Sun1sfGFP mice (without Cre), were

used as control in Ext. Data Fig 1.

Fig 1 states that cervical, thoracic and lumbar regions were processed separately. However, it is not clear - in all subsequent figures (except Figure 6) and manuscript text - whether the RNA-Seq data shown come from all these regions. This is very important especially for skeletal MN clustering because different subtypes of these neurons occupy distinct regions along the spinal cord. Also, there is an important conclusion hidden in legend of Ext. Data Fig 3 (larger numbers of skeletal MNs were detected in cervical and lumbar regions compared to thoracic).

Since the sequencing was done on cervical, thoracic and lumbar motor neurons, why there is no attempt to connect alpha, beta and gamma motor neurons with, for example, LMC and MMC populations. Based on the authors' data, is there any evidence for MN columns or pools in adults? Post-natal pool markers have been reported for digit-innervating MNs (Mendelsohn et al., *Neuron*, 2017). About 50 MN pools are estimated at the limb level in mice. Do the authors believe that this diversity is somehow "erased" in the adult based on their molecular atlas?

What are the technical limitations of the current study? A paragraph on this in Discussion could help the reader. Have the authors sequenced enough cells to categorize skeletal MNs? Was the nucleus size taken into account during sorting? Does their analysis underestimate the diversity of cholinergic motor neurons?

To unambiguously assign identity to cluster S1, why don't the authors use previously published markers for gamma MNs by Rosenberg et al (2018) ? A list of markers for alpha and gamma MNs at postnatal stages was identified in 2018 by Rosenberg et al.

Clustering scRNA-Seq data can result in different outcomes depending on the parameters used. Can the authors comment on plausible reasons for the following discrepancy? *Tns1* is shown as a gamma MN marker by Rosenberg et al, while the current study shows that *Tns1* marks all 3 classes of skeletal MNs.

One of the most intriguing findings is that distinct clusters of visceral MNs are found in different regions (cervical, thoracic, and lumbar). However, the excitement for this discovery is decreased by the limited validation with ISH (*Sst*, *Bnc2*) of the putative visceral MN subtypes at different locations of the spinal cord. Have the authors confirmed that *Bnc2*+ visceral MNs are not present in thoracic and lumbar regions? A similar question goes for *Sst*+ visceral MNs. Further validation of additional markers by ISH at different spinal cord regions will strengthen the conclusion of the visceral MN "body map" mentioned in Discussion (line 330).

In Discussion, it is stated that this work aligns well with a concurrent study (Plum et al., posted in bioRxiv on March of 2020). It is comforting to see that two independent studies conducted in different labs reached largely similar conclusions. However, it is puzzling that the genes chosen for ISH validation in this study (*Fbn2*, *Tns1*, *Sv2b*, *Glis3*, *Nrp2*, *Rreb1*, *Plekhg1*, *Kcnq5*, *Piezo2*, *Set*, *Penk*) are also highlighted by Plum et al. If the authors consulted Plum et al for choosing these genes, they should acknowledge it in text. In the current version, the Plum et al paper is cited once when the authors discuss their findings in interneurons.

In lines 66-68, the authors state that the data provide insights into the normal physiological functions and susceptibility to disease. However, I am not convinced these claims are supported by the data. For example, silencing or killing of newly discovered neuron types to decipher their function has not been performed, like in similar studies (Okaty et al 2015), and this is anyway hard to do in the spinal cord. Moreover, whether some of these newly identified neuron types are selectively vulnerable to motor neuron disease is not evaluated.

Minor comments

Paragraph spanning 189-196 belongs to Discussion. It should also cite Rosenberg et al (2018) and Plum et al.(2020).

Similarly, paragraph (lines 257-263) probably fits best in Discussion.

line 80: the authors generated a new mouse strain by crossing two available mouse lines. They did not "engineer" new mice.

line 96: the conclusion is vague. A direct comparison with previous studies is suggested.

Is Fbn2 labeling visceral MNs in all spinal levels? Only thoracic is shown in 2e.

Re-clustering is an essential method used in this paper and many other sc-RNA-Seq studies. But not enough detail is provided in Methods on how re-clustering was done.

Reclustering of skeletal MNs revealed 8 clusters. This is somewhat contradictory to the statement in Abstract about limited heterogeneity of these cells. It would help the reader if more information on how the re-clustering of so "tightly clustered" cells was done. What was the basis for selectively re-clustering skeletal and visceral MNs, but not cholinergic interneurons?

The authors find Piezo2 to be expressed in cholinergic interneurons and visceral MNs (V2, V3, V7, V8). Plum et al describe Piezo2 is also expressed in alpha MNs. Given the surprising expression of this molecule in MNs, it would be good to clarify whether the authors did not find Piezo2 in alpha MNs.

Lines 222-223: Quantification should be provided to support the claim that all Rreb1+ alpha MNs express Sv2a, but not Chodl.

Reviewer #3:

Remarks to the Author:

The present manuscript by Alkaslasi et al. aims at unraveling the diversity of cholinergic neurons on the transcriptional and single cell level in order to define new marker for specific cholinergic neuronal subtypes. By labelling cholinergic neuron nuclei using Chat-IRES-Cre::Sun1/SfGFP double transgenic animals, Chat-expressing nuclei were enriched for subsequent single nuclei RNA sequencing. Differential gene expression analysis revealed 19 distinct clusters of cholinergic neurons in the spinal cord of 8 weeks old mice. The clusters were assigned to the three main classes by correlating the anatomical localization of the nuclei with already established markers for the three main classes using in situ hybridization probes against RNA molecules (IHC). Additionally, a set of uniquely expressed genes allowing the specific allocation of each cluster to one of the three main classes was identified. By this approach, the authors identified 3 subtypes of skeletal motor, 8 of visceral motor and 8 interneuron subtypes and their distribution along the spinal cord. The subtype-specificity of selected markers was shown by extensive IHC stainings. Finally, the subtypes were re-clustered within their respective main class showing an even higher transcriptional diversity. The distinct skeletal motor neurons cluster were co-related to the different functional types of skeletal motor neurons, being alpha, beta and gamma motor neurons and within the alpha group fast and slow neurons by IHC stainings using established markers for the respective functional types.

The reviewer is convinced by the overall quality of the paper. The unbiased approach of single nuclei transcriptional analysis combined with extensive IHC stainings clearly demonstrates the validity of the newly identified marker genes as a descriptive tool to differentiate cholinergic

neuron classes. Furthermore, selected markers were nicely shown to specifically label proposed cholinergic subtypes in the spinal cord.

However, the reviewer feels that while the present data sets represent an important and essential basis for a deeper understanding of the diversity of cholinergic neurons with potential implications for the pathogenesis of degenerative motor neuron diseases, the manuscript would benefit from a more extensive and complete investigation of the hitherto findings. In detail, 1) a correlation of the transcriptional findings to protein expression, 2) the functional assignment of skeletal MN subgroups, 3) the usefulness of the identified markers for distant labeling in peripheral end organs would strongly improve the significance of the single cell analysis (for details see major points). In addition, the manuscript would profit from any information about the dynamics of the identified transcriptional cholinergic neuron subsets. This could include the analysis of temporal changes, e.g. with regard to development or ageing, or a comparison of the presented data sets with a diseases context, e.g. by taking advantage of a motor neuron disease model. Indeed the authors suggest that subpopulations of cholinergic motor neurons may be more susceptible to degeneration in respective diseases.

Major points:

1) Immunohistochemical analysis for the major identified subpopulation markers should be included in order to obtain any information on the protein expression of the respective transcripts. This would be highly useful for any further studies building on the present manuscript.

2) Labeling was restricted to the spinal cord while it remains to show that these markers can also be used for distant stainings in the peripheral end organs for skeletal MNs and at pre-ganglionic sites for visceral MNs

3) In line, snRNA Seq is prone to enrich for nuclear related/specific transcripts. Differentiating between genes with specific nuclear functions and cytosolic genes is an important information regarding the usefulness of the novel markers in the periphery outside the spinal cord

4) While the diversity of visceral and interneurons was tackled solely on the descriptive level, skeletal motor neuron clusters were suggested to correlate with functional classes of motor neurons based on the expression of a set of published marker genes. Although stainings revealed a colocalization, the overall expression of e.g. Chodl is restricted to a small subset of nuclei in the alpha A2 subgroup or other intended specific markers were expressed in all skeletal MN subtypes (Esrrg and Gfra1, Fig. 3a). Therefore, the functional assignment of skeletal MNs subgroups is vague and remains to be proven by further determining the innervated muscle fiber types of the respective skeletal MN subgroups.

5) In this regard:

156-168: Assignment of S1 to gamma and S3 to beta MNs is elusive. Esrrg and Gfra1 are expressed by all skeletal MNs indicating these as rather general markers. IHC co-staining with new marker and marker for muscle fiber types in muscle tissue or whole cell labeling and innervation tracking, or similar would be necessary to support the functional classification. > Sv2b and Rbfox3 co-staining may suggest alpha MN identity

212-233: Classification of transcriptional subgroups within the skeletal MNs is reasonable. However, the physiological classification based solely on gene expression without functional correlation is much too vague (see comment above)

> Fig 4d: Chodl is only expressed in a small number of nuclei within skeletal MN cluster A2 making it difficult to judge the validity as reliable marker for functional classification of alpha MNs

> Fig 4 g,h: CoStaining of Sv2a with Chodl and Rreb1 needed to support proposed result

6) Selection criteria for regulated genes between clusters lack the information for normalization. Was the log-fold change determined by the average expression of the clusters against all sequenced nuclei or normalized to the expression of housekeeper which were uniform over all nuclei?

Minor points:

- Many IHC images have different scales within one figure making a comparison difficult especially

when the authors are writing about the size of nerves

- Explanation for violin plot generation is missing. Number of included cells per plot per cluster is essential for comparison of expression data
- Figure 2e: More neurons left from Box 3 are ChAT positive. What neurons are those?
- Fig2 e,f: Single staining images for all staining in all boxes needed for proper validation of the specificity of the marker
- Line 195-196: Sv2b is specific for alpha MNs, but previously known Rbfox3 (NeuN) too according to Fig 3a. What is the advance?

REVIEWER COMMENTS

Reviewer #1 (Remarks to the Author):

This manuscript by Alkaslasi et al. describes a single nuclear RNA sequencing approach used on the adult spinal cord to identify cholinergic (Chat) classes of interneurons and motor neurons based on their transcriptional profile. The authors use a neuronal and Chat specific fluorescent sorting and to optimize their harvesting of Chat+ cells from the cord. They first identify 31 clusters that are neuronal – with 13 co-expressing either Slc176a or Gad1 – and then subtract the Slc176a or Gad1 clusters to end up with 19 (??) ‘pure’ Chat+ cluster. Through data mining and using previous genes in Chat+ spinal neurons as entry points, they categorize clusters as visceral preganglionic motor neurons, interneurons, gamma, beta and alpha-motor neurons (the later with fast and slow motor neuron groups). They conclude that their data provide a new and comprehensive categorization of Chat+ neurons in the spinal cord that may be used to perform targeted functional studies.

The study is very well carried out as a transcriptional analysis and because it use a single nuclear RNA sequencing approach it is likely that it has been able to capture more cells and therefore presumably more transcripts than the fair number of previous studies using single cell sequencing or bulk sequencing (especially of the large alpha-MNs that are fragile for FACS). From that point of view the study could provide important new information. My enthusiasm for the study is, however, dampened considerably because of lack of functional confirmation of any of the clusters. This leads to a number of very strong claims that are unsubstantiated (possible misleading?) and need direct confirmation. Essentially the categorization into the three main categories (visceral motor neurons, ventral motor neurons, and interneurons) are based on unique markers identified in previous studies and then from there a subcategorization is made. This means that the categorization is biased beforehand. There are no independent means of knowing how many cells that are captured by the analysis. Such an estimate could have been obtained by retrogradely labelling motor neurons from the ventral roots combined with expression analysis. Now the only verification is the location in the cord and the soma size. But this is not enough to do the functional/anatomical verification. For example, retrograde labelling from ventral root of cells in the cervical cord is required to support the claim that the author have found a hitherto unrecognized population of visceral cells in that region. The suggested experiment is complex and would provide little information other than that motor neurons in the lateral horn of the cervical spinal cord exist. We used a simpler and even more diagnostic alternative to address this issue and now provide a cleared view of Chat-Cre::Sun1-GFP+ nuclei in the cervical spinal cord region demonstrating the existence of a lateral horn pool of visceral cells in the caudal region of cervical cord (Supplementary Figure 13a-c). It is a direct extension of the lateral column that exists in the thoracic spinal cord. We also have performed in situ hybridization in sections of this cervical level demonstrating the existence of these neurons in the lateral column that express *Chat* and *Fbn2*, the visceral marker we identified (Supplementary Figures 12a and 13d-h and Figure 6f). Similarly, to verify the slow and fast motor neuron markers indeed are markers or these pools of motor neurons retrograde labelling from ‘fast’ and ‘slow’ hindlimb muscles are required to directly confirm the finding which will also give information about whether motor neurons are flexor or extensors. The referee makes a good point. We carried out the experiments as suggested and discovered that our preliminary assignment of fast and slow motor neurons based on previous literature was wrong. With the additional sequencing requested by other referees, we now distinguish 8 subtypes of alpha MNs. Using retrograde labeling from specific muscle

types, we show that 2 types selectively innervate digit versus diaphragm (Figure 4).

Also to show that a gamma motor neuron is a gamma motor neuron it is not enough to look at size – there are many indirect criteria that need to be fulfilled to say that a cell in the ventral horn is a presumptive gamma motor neuron (see Friese et al. 2009; Shneider et al. 2009) some of which the authors should use for identification (retrograde labelling, lack of Vglut1 synapses, lack of C-boutons). Admittedly this is extra work but, in my opinion, needed to make a claim about identity.

We provide new data to validate our predicted alpha and gamma motor neuron markers by combining in situ hybridization with pre-imaging of CHAT expression from Chat-Cre::Ai14 mouse tissue that clearly highlights C-boutons onto CHAT-positive cell bodies. We show that alpha motor neuron cell bodies surrounded by C-boutons express our predicted alpha motor marker but not the gamma marker, and that gamma neurons lacking C-boutons express our predicted gamma marker, but not the alpha marker (Figure 3e). We also show that putative gamma neurons also lack VGLUT1+ synapses while SV2B+ alpha MNs cell bodies exhibit VGLUT1 synapses (Extended Data Figure 9a).

To identify a gamma motor neuron directly can only be done electrophysiologically – which needs to be recognized. We agree with the reviewer that the definitive identification of a gamma neuron would be by electrophysiology and we acknowledge this (line X, page 5-6).

The categorization as a beta motor neuron is problematic. Very few publications talk about beta-motor neurons. This is because it is extremely difficult to show a motor neuron is a beta motor neuron. It is a requirement to show that the motor neurons anatomically innervate but striatal musculature and muscle spindles (alternatively show that physiologically). The authors are not doing any test to show that and can therefore they cannot claim that they have found markers for beta-motor neurons.

We agree with the reviewer and have toned down our claims throughout the text (abstract, results and discussion). We clearly observe the existence of a third population of skeletal motor neurons that is transcriptomically distinct from alpha and gamma, yet more closely related to alpha, that we hypothesize are beta motor neurons. Indeed, the large number of these neurons, their small size, and lack of C-boutons (new data), but closer transcriptomic similarity to alpha MNs all support this hypothesis.

In conclusion: the study is potentially very interesting showing new markers for Chat+ spinal cells. However, the lack of independent verification of cell type categorization is a major weakness of the study which unfortunately make the strong claims about new cell markers unsubstantiated. This is unfortunate because the study could have been taken so much further. Without these experiments the study remains descriptive and suggestive with regard to cell type categorization. We thank the referee for constructive suggestions that we agree have strengthened the manuscript significantly.

Reviewer #2 (Remarks to the Author):

The work by Alkaslasi, Piccus et al. provides a molecular atlas of cholinergic neurons found in the adult mouse spinal cord. The authors used a genetic approach to permanently mark and isolate nuclei from this rare population of neurons. This by itself is an important achievement because cholinergic motor neurons are notoriously difficult to isolate from adult spinal cords. Next, they performed single-nucleus RNA-Seq on samples collected from cervical, thoracic and lumbar regions. They identified 19 distinct clusters. Based on previous markers and computational analysis, they break down these to interneurons (I1-I8), visceral MNs (V1-V8) and skeletal MNs (S1-S3). Through re-clustering, they identify substantial diversity for visceral and skeletal MNs and suggest novel markers for these cell types. Collectively, the

findings provide a comprehensive transcriptomic description of spinal cholinergic neurons. This work is of interest to the field of motor neuron biology and disease. Most conclusions are supported by evidence, unless otherwise noted (see Major Concerns). However, I am not convinced that it provides a conceptual advance to influence thinking in the field. In other words, a transcriptomic description of cholinergic neurons in the adult spinal cord is certainly valuable, but to advance the field the authors should use their molecular atlas to answer an important biological question and provide new insights, especially in light of previous reports hinting molecular diversity in spinal cord neurons. Comparing this work to others (Okaty et al., *Neuron* 2015; Sathyamurthy et al., *Cell Reports.*, 2018; Tran et al., *Neuron*, 2019), it seems that the current manuscript, after establishing a molecular atlas, does not take the next crucial step to answer a biological question. Hence, the current version of this manuscript, in my humble opinion, does not make a conceptual contribution that meets the standards of *Nature Communications*.

Major concerns

Establishing a molecular resource and then “using” it to provide new knowledge is what distinguishes a descriptive study from a substantial contribution. Since the major discovery of this paper is the identification of a dozen visceral MN subtypes, the authors could use, for example, an ALS mouse model and test whether these subtypes are differentially susceptible to disease, similar to a study on retina ganglion cells (Tran et al., *Neuron*, 2019). Evidence that some visceral MNs may be affected in ALS has been reported previously <https://onlinelibrary.wiley.com/doi/abs/10.1002/mus.24457>. Moreover, cholinergic interneurons are also affected in a mouse model of ALS. <https://www.ncbi.nlm.nih.gov/pmc/articles/PMC3607155/> Testing whether the identified subtypes of visceral MNs and/or cholinergic interneurons are differentially susceptible to disease would dramatically strengthen this work and its conceptual advance. *We agree with the reviewer that this study is more of a resource for the community and will spur research. Moreover, it completely rewrites what is known about cholinergic neurons in the spinal cord. Addition of retrograde labeling, requested by reviewers, further adds to our characterization of these neurons. Adding disease models would be many months of work even without current restrictions due to the COVID-19 pandemic. For example, a proper study of these neurons in a disease model would involve crossing the disease model to the Chat-Cre::Sun1GFP mice and then harvesting them at multiple time points to perform a longitudinal analysis. We think it would be wrong to leave this work unpublished in the absence of sequencing in a disease model. Our goal in this manuscript is to describe and validate the different classes of neurons in healthy adult spinal cord. It lays the groundwork for future studies to examine how the identities of these neurons change in disease, but that is outside the scope of this paper.*

It is curious that from the 14,738 nuclei sequenced, only 6,941 were expressing ChAT. The authors suggest that the large number of excitatory and inhibitory neurons captured likely reflects that ChAT-IRES-Cre was expressed in these cells or their lineage. However, their genetic method permanently marked cholinergic cells in expected locations (lines 83-84). Fig. 1 clearly shows that the vast majority of GFP+ nuclei are indeed in these expected locations. Only a minority of GFP+ cells is outside these locations. Hence, an alternative explanation should be provided to clarify how ~8,000 of the 14,738 nuclei turned out to be excitatory and inhibitory neurons. A suggestion would be to test whether the ROSA-Sun1sfGFP mice (without Cre) show any baseline recombination events. Of note, wild-type mice, not ROSA-Sun1sfGFP mice (without Cre), were used as control in Ext. Data Fig 1. *At the resolution we had initially provided in Fig.1, it was difficult to see the GFP+ cells outside of expected locations. We now provide higher resolution views clearly showing GFP+ nuclei outside of the ventral and lateral horns, and the intermediate zone (Supplementary Figure 3b-c). We also now provide the requested control spinal*

cord shown at different levels (C, T, L) that show no GFP signal is detected in these Cre-negative mice (Supplementary Figure 3a). Therefore, our explanation is supported by data and the alternative suggested by the reviewer is ruled out.

Fig 1 states that cervical, thoracic and lumbar regions were processed separately. However, it is not clear - in all subsequent figures (except Figure 6) and manuscript text - whether the RNA-Seq data shown come from all these regions. This is very important especially for skeletal MN clustering because different subtypes of these neurons occupy distinct regions along the spinal cord. Also, there is an important conclusion hidden in legend of Ext. Data Fig 3 (larger numbers of skeletal MNs were detected in cervical and lumbar regions compared to thoracic). We thank the reviewer for pointing out that we did not make full use of the way we performed the sequencing. We clearly explain our strategy to separate the different spinal cord regions in the initial results section. We have also expanded the regional analysis (C, T, L/S) to the skeletal motor neurons (Figure 4) and the interneurons (Supplementary Figure 11). Combinatorial clustering was performed on the entire dataset but the level of origin can be identified afterwards. We have indeed uncovered interesting differences between the C, T, and L/S regions for both the skeletal and visceral MNs. We agree this strengthens our conclusions.

Since the sequencing was done on cervical, thoracic and lumbar motor neurons, why there is no attempt to connect alpha, beta and gamma motor neurons with, for example, LMC and MMC populations. To examine this, we retrogradely labeled motor neurons that project to the lumbar extensors of the spine (axial muscles; MMC) as well as several limb muscles (LMC), but we did not observe strong expression of particular markers of alpha MN subtypes in these cells (Figure 4; Supplementary Fig 10).

Based on the authors' data, is there any evidence for MN columns or pools in adults? Post-natal pool markers have been reported for digit-innervating MNs (Mendelsohn et al., Neuron, 2017). We do see *Cpne4+* alpha MNs enriched in digit-innervating MNs (Figure 4b, e, g), in agreement with Mendelsohn et al., and a different set of neurons innervating the phrenic nerve (Figure 4). The referee's suggestion of considering different levels separately also supports this view.

About 50 MN pools are estimated at the limb level in mice. Do the authors believe that this diversity is somehow "erased" in the adult based on their molecular atlas? We agree this is the most likely explanation and address this in our results and discussion sections.

What are the technical limitations of the current study? A paragraph on this in Discussion could help the reader. Have the authors sequenced enough cells to categorize skeletal MNs? Was the nucleus size taken into account during sorting? Does their analysis underestimate the diversity of cholinergic motor neurons? To address the concern that we did not capture sufficient numbers of cholinergic neurons to properly discern their diversity, we have sequenced additional neurons (sequenced *Chat+* nuclei were increased from 6,941 to 16,042), more than doubling the n we have analyzed. Reassuringly, the overall clustering pattern was conserved, demonstrating that the initial dataset was reliable and reproducible. We have added a sentence in the discussion to reflect this: when we analyze a subset or double the number of nuclei analyzed, it does not change the major conclusions, so we have reached a point where further sequencing is of diminishing returns. Previous scRNA seq of these neurons has identified a few dozen neurons. This study increases it by orders of magnitude.

We did not take into account the nucleus size during sorting.

We have added a description of technical limitations to the discussion.

To unambiguously assign identity to cluster S1, why don't the authors use previously published markers

for gamma MNs by Rosenberg et al (2018) ? A list of markers for alpha and gamma MNs at postnatal stages was identified in 2018 by Rosenberg et al. Clustering scRNA-Seq data can result in different outcomes depending on the parameters used. Can the authors comment on plausible reasons for the following discrepancy? Tns1 is shown as a gamma MN marker by Rosenberg et al, while the current study shows that Tns1 marks all 3 classes of skeletal MNs. We did not refer to the markers defined in Rosenberg et al (2018) because they misassigned identities. This is likely because they only identified 177 cholinergic cells and overlooked the autonomic motor neurons. Our analysis is much more comprehensive, including 16,042 cholinergic cells, and we now provide independent evidence supporting our conclusions.

One of the most intriguing findings is that distinct clusters of visceral MNs are found in different regions (cervical, thoracic, and lumbar). However, the excitement for this discovery is decreased by the limited validation with ISH (Sst, Bnc2) of the putative visceral MN subtypes at different locations of the spinal cord. Have the authors confirmed that Bnc2+ visceral MNs are not present in thoracic and lumbar regions? A similar question goes for Sst+ visceral MNs. Further validation of additional markers by ISH at different spinal cord regions will strengthen the conclusion of the visceral MN “body map” mentioned in Discussion (line 330). In our revision we have investigated and quantified regional-specific markers at all 4 levels (Figure 6 and Supplementary Figure 12). Our new data fully support the conclusions from the sequencing.

In Discussion, it is stated that this work aligns well with a concurrent study (Plum et al., posted in bioRxiv on March of 2020). It is comforting to see that two independent studies conducted in different labs reached largely similar conclusions. However, it is puzzling that the genes chosen for ISH validation in this study (Fbn2, Tns1, Sv2b, Glis3, Nrp2, Rreb1, Plekhg1, Kcnq5, Piezo2, Set, Penk) are also highlighted by Plum et al. If the authors consulted Plum et al for choosing these genes, they should acknowledge it in text. In the current version, the Plum et al paper is cited once when the authors discuss their findings in interneurons. These markers came strictly out of our sequencing data analyses (see Supplementary Table 2). We did not consult with Blum et al. in order to find them. The agreement of our data with theirs fully supports the validity of our different sequencing approaches.

In lines 66-68, the authors state that the data provide insights into the normal physiological functions and susceptibility to disease. However, I am not convinced these claims are supported by the data. For example, silencing or killing of newly discovered neuron types to decipher their function has not been performed, like in similar studies (Okaty et al 2015), and this is anyway hard to do in the spinal cord. Moreover, whether some of these newly identified neuron types are selectively vulnerable to motor neuron disease is not evaluated. We have deleted the phrase “susceptibility to disease” and support the “normal physiological functions” with new data as requested by the referees.

Minor comments

Paragraph spanning 189-196 belongs to Discussion. It should also cite Rosenberg et al (2018) and Plum et al.(2020). Similarly, paragraph (lines 257-263) probably fits best in Discussion. Both these references are cited in the discussion, but new results mean these sections belong in the Results.

line 80: the authors generated a new mouse strain by crossing two available mouse lines. They did not “engineer” new mice. We agree.

line 96: the conclusion is vague. A direct comparison with previous studies is suggested. We have added

the requested details.

Is *Fbn2* labeling visceral MNs in all spinal levels? Only thoracic is shown in 2e. Yes, we have performed in situ hybridization for *Chat* and *Fbn2* in all spinal levels for our quantifications shown in Figure 6e, and we observe *Fbn2* in many *Chat*+ neurons of the lateral horn at all levels (Extended Data Figures 9 and 13).

Re-clustering is an essential method used in this paper and many other sc-RNA-Seq studies. But not enough detail is provided in Methods on how re-clustering was done. An explanation of the re-clustering has been added to this methods section (page 13).

Reclustering of skeletal MNs revealed 8 clusters. This is somewhat contradictory to the statement in Abstract about limited heterogeneity of these cells. It would help the reader if more information on how the re-clustering of so “tightly clustered” cells was done. What was the basis for selectively re-clustering skeletal and visceral MNs, but not cholinergic interneurons? We agree that for completions’ sake, we should provide as much detail for the cholinergic interneuron clustering as the other main types. We have now performed re-clustering for cholinergic interneurons as well; this is shown in Extended Data Figure 11a.

The authors find *Piezo2* to be expressed in cholinergic interneurons and visceral MNs (V2, V3, V7, V8). Plum et al describe *Piezo2* is also expressed in alpha MNs. Given the surprising expression of this molecule in MNs, it would be good to clarify whether the authors did not find *Piezo2* in alpha MNs. We detect much higher expression levels of *Piezo2* in some visceral MN clusters (V2, V3, V4, V5, V7) as well as in previously undescribed non-motor neuron types (clusters I1, I7 and I8). We also detect low levels of *Piezo2* in a very small subset of MNs (alpha MN subcluster 6, and skeletal subcluster 9, which is a gamma MN subtype). Now that Blum et al.’s data are accessible, we see the same pattern in their dataset, so we are not sure why they highlight the expression of *Piezo2* in alpha MNs.

Lines 222-223: Quantification should be provided to support the claim that all *Rreb1*+ alpha MNs express *Sv2a*, but not *Chodl*.

Based on our new data, we no longer concentrate on the *Chodl* vs not-*Chodl* distinction since it is not supported by our new results. This is one example of where the increased number of neurons sequenced was helpful in performing a more accurate analysis and the retrograde tracing studies recommended by the reviewers were especially useful.

Reviewer #3 (Remarks to the Author):

The present manuscript by Alkaslasi et al. aims at unraveling the diversity of cholinergic neurons on the transcriptional and single cell level in order to define new marker for specific cholinergic neuronal subtypes. By labelling cholinergic neuron nuclei using *Chat*-IRES-Cre::Sun1/SfGFP double transgenic animals, *Chat*-expressing nuclei were enriched for subsequent single nuclei RNA sequencing. Differential gene expression analysis revealed 19 distinct clusters of cholinergic neurons in the spinal cord of 8 weeks old mice. The clusters were assigned to the three main classes by correlating the anatomical localization of the nuclei with already established markers for the three main classes using in situ hybridization probes against RNA molecules (IHC). Additionally, a set of uniquely expressed genes allowing the specific allocation of each cluster to one of the three main classes was identified. By this approach, the authors identified 3 subtypes of skeletal motor, 8 of visceral motor and 8 interneuron subtypes and their distribution along the spinal cord. The subtype-specificity of

selected markers was shown by extensive IHC stainings. Finally, the subtypes were re-clustered within their respective main class showing an even higher transcriptional diversity. The distinct skeletal motor neurons cluster were co-related to the different functional types of skeletal motor neurons, being alpha, beta and gamma motor neurons and within the alpha group fast and slow neurons by IHC stainings using established markers for the respective functional types.

The reviewer is convinced by the overall quality of the paper. The unbiased approach of single nuclei transcriptional analysis combined with extensive IHC stainings clearly demonstrates the validity of the newly identified marker genes as a descriptive tool to differentiate cholinergic neuron classes. Furthermore, selected markers were nicely shown to specifically label proposed cholinergic subtypes in the spinal cord.

However, the reviewer feels that while the present data sets represent an important and essential basis for a deeper understanding of the diversity of cholinergic neurons with potential implications for the pathogenesis of degenerative motor neuron diseases, the manuscript would benefit from a more extensive and complete investigation of the hitherto findings. In detail, 1) a correlation of the transcriptional findings to protein expression, 2) the functional assignment of skeletal MN subgroups, 3) the usefulness of the identified markers for distant labeling in peripheral end organs would strongly improve the significance of the single cell analysis (for details see major points).

In addition, the manuscript would profit from any information about the dynamics of the identified transcriptional cholinergic neuron subsets. This could include the analysis of temporal changes, e.g. with regard to development or ageing, or a comparison of the presented data sets with a diseases context, e.g. by taking advantage of a motor neuron disease model. Indeed the authors suggest that subpopulations of cholinergic motor neurons may be more susceptible to degeneration in respective diseases.

The single nucleus sequencing allows us to define *combinations of markers* that specifically label certain neuron classes and can be detected by multiplexed in situ hybridization. Thus, the goal of the single cell or single nucleus sequencing is to use the full transcriptome to define neuronal classes and identify diagnostic marker combinations. To make this clear to a general audience, we have elaborated on this point at the beginning of the results.

Major points:

1) Immunohistochemical analysis for the major identified subpopulation markers should be included in order to obtain any information on the protein expression of the respective transcripts. This would be highly useful for any further studies building on the present manuscript. For this revision, we tested multiple candidate antibodies. We succeeded in identifying a reliable antibody against SV2B, our newly defined alpha motor neuron marker, and include examples of SV2B staining in the spinal cord as well as the periphery (Extended Data Figure 9). This antibody (Synaptic Systems #119102) was validated by the manufacturer for its specificity by lack of signal detection in knockout mouse tissue. Other antibodies were less convincing, potentially because of cross-reactivity, expression localized to regions of cells outside the spinal cord, and expression in cells outside of the cholinergic cells of interest, etc...

2) Labeling was restricted to the spinal cord while it remains to show that these markers can also be used for distant stainings in the peripheral end organs for skeletal MNs and at pre-ganglionic sites for visceral MNs. The antibody against SV2B described above produced good quality pre-synaptic staining in skeletal muscle (Extended Data Figure 9), so we were able to show the validity of this particular marker

in a peripheral tissue, for a protein marker that would be expected to be localized to presynaptic terminals.

3) In line, snRNA Seq is prone to enrich for nuclear related/specific transcripts. Differentiating between genes with specific nuclear functions and cytosolic genes is an important information regarding the usefulness of the novel markers in the periphery outside the spinal cord. We agree with the reviewer that the protein function and localization is critical in determining whether a particular marker can be used in the periphery. However, the main goal of our work was to determine genetic markers that can distinguish between neuronal cell classes. These can be used as we have shown, by multiplexed in situ hybridization in the spinal cord. In future, they can also serve as a guide for making useful Cre driver lines and viral vectors, which would be another way of labeling specific subsets of neurons without having to rely on antibody staining. We have now added a caveat about nuclear sequencing to the discussion.

4) While the diversity of visceral and interneurons was tackled solely on the descriptive level, skeletal motor neuron clusters were suggested to correlate with functional classes of motor neurons based on the expression of a set of published marker genes. Although stainings revealed a colocalization, the overall expression of e.g. *Chodl* is restricted to a small subset of nuclei in the alpha A2 subgroup or other intended specific markers were expressed in all skeletal MN subtypes (*Esrrg* and *Gfra1*, Fig. 3a). Therefore, the functional assignment of skeletal MNs subgroups is vague and remains to be proven by further determining the innervated muscle fiber types of the respective skeletal MN subgroups. We have taken steps to validate the alpha MN markers as being expressed in cell bodies that receive C-boutons and VGLUT1+ synapses, and gamma MN markers in cell bodies that lack C bouton and VGLUT1+ synapse innervation (Figure 3e). We further categorized classes of alpha MNs according to the muscle groups they innervate using retrograde labeling (Figure 4; see Reviewer 2).

5) In this regard:

156-168: Assignment of S1 to gamma and S3 to beta MNs is elusive. *Esrrg* and *Gfra1* are expressed by all skeletal MNs indicating these as rather general markers. IHC co-staining with new marker and marker for muscle fiber types in muscle tissue or whole cell labeling and innervation tracking, or similar would be necessary to support the functional classification. Our data demonstrated that *Esrrg* and *Gfra1* are not specific markers for skeletal MN classes, contrary to previous reports. We agree that the classification of the beta MN group is elusive in the absence of innervation tracking and physiological validation. (See also reviewer 1.) We have therefore rephrased our results section regarding the putative beta MNs and toned down this claim throughout the manuscript. We observe a 3rd class of skeletal MN that is clearly distinct from alpha and gamma, and we hypothesize it corresponds to beta. In situ hybridization using a novel marker (*Gpr149*) (Figure 3e and Supplementary Figure 8b, g) validates the existence of this neuron type: a small diameter motor neuron without prominent C-boutons.

> *Sv2b* and *Rbfox3* co-staining may suggest alpha MN identity To address this point, we now provide in situ hybridization showing co-expression of *Rbfox3*, *Sv2b*, and *Stk32a* in skeletal alpha motor neurons, but not in gamma neurons (Figure 3d).

212-233: Classification of transcriptional subgroups within the skeletal MNs is reasonable. However, the physiological classification based solely on gene expression without functional correlation is much too vague (see comment above) Our extended sequencing data and retrograde tracing demonstrate that the reviewer was right. See also comments to Reviewer 2.

> Fig 4d: *Chodl* is only expressed in a small number of nuclei within skeletal MN cluster A2 making it difficult to judge the validity as reliable marker for functional classification of alpha MNs. Based on the additional sequencing data, the alpha MNs subcluster into 8 groups instead of just 2, and *Chodl* is only expressed in a small proportion of cells, therefore we did not pursue its characterization (Figure 4). See also comments to Reviewer 2 and response to major point 4.

> Fig 4 g,h: CoStaining of Sv2a with *Chodl* and *Rreb1* needed to support proposed result. As noted above, our new sequencing data have shifted the focus away from *Rreb1* as a marker of a true subclass of alpha MN. *Rreb1* has widespread expression within alpha MN subclusters (Extended Data Figure 14).

6) Selection criteria for regulated genes between clusters lack the information for normalization. Was the log-fold change determined by the average expression of the clusters against all sequenced nuclei or normalized to the expression of housekeeper which were uniform over all nuclei? All our analyses were performed using standard methods for single cell analysis that have been well validated. Specifically, we used the *FindAllMarkers* function in *Seurat*. This is described in the Methods (Butler et al. 2018).

Minor points:

- Many IHC images have different scales within one figure making a comparison difficult especially when the authors are writing about the size of nerves. We have avoided changing scales where possible.
- Explanation for violin plot generation is missing. Number of included cells per plot per cluster is essential for comparison of expression data. We have added this information to the methods and the number of cells analysed is included in Supplementary Table 1.
- Figure 2e: More neurons left from Box 3 are ChAT positive. What neurons are those? The additional *Chat+* neurons to the left of Box 3 are likely interneuron types. It is hard to assign an identity without using additional probes.
- Fig2 e,f: Single staining images for all staining in all boxes needed for proper validation of the specificity of the marker. These are provided in Supplementary Figure 6c, d. We have also highlighted this in the figure legend for Figure 2.
- Line 195-196: Sv2b is specific for alpha MNs, but previously known *Rbfox3* (NeuN) too according to Fig 3a. What is the advance? *Sv2b* and *Stk32a* are both specific markers for alpha MNs that we have defined from our sequencing data. The advantage of these markers is that their expression is much more restricted than that *Rbfox3* among all other spinal cord neurons. For example, they could be used to generate useful Cre driver lines.

Reviewers' Comments:

Reviewer #1:

Remarks to the Author:

The authors have gone a fair distance in their revision to meet the critique that I raised in my previous report. In particular they have provided more direct evidence to verify that the groups of somatic and visceral motor neurons they molecularly characterize belong to the subclasses they suggest. However, I still have a problem with the claims about the beta motor neurons. There is no direct evidence for that characterization but despite that the authors keeps calling them beta-motor neurons in the text and figure legends. This is not appropriate and they should change it to putative beta motor neurons or in some other way with the naming indicate that they have no direct evidence for the nature of this group of MNs.

Reviewer #2:

Remarks to the Author:

The revised manuscript has addressed all my concerns and is very much improved.

Reviewer #3:

Remarks to the Author:

In the revised version of the manuscript, the authors have substantially improved the quality of the manuscript and have addressed the main points of the reviewer. This refers to the 1) identification of specific novel markers for the individual MN subtypes, 2) the retrograde labelling of different muscles and 3) and improvement of the discussion with regard to the interpretation of their findings.

In summary, the present study represents an important basis for a deeper understanding of the diversity of cholinergic neurons and the reviewer has no further concerns.

Reviewer #1 (Remarks to the Author):

The authors have gone a fair distance in their revision to meet the critique that I raised in my previous report. In particular they have provided more direct evidence to verify that the groups of somatic and visceral motor neurons they molecularly characterize belong to the subclasses they suggest. However, I still have a problem with the claims about the beta motor neurons. There is no direct evidence for that characterization but despite that the authors keeps calling them beta-motor neurons in the text and figure legends. This is not appropriate and they should change it to putative beta motor neurons or in some other way with the naming indicate that they have no direct evidence for the nature of this group of MNs.

Response to Reviewer #1

We have taken the reviewer's suggestion, and now refer to this class of neuron as a third type of skeletal motor neuron, distinct from alpha and gamma, which may correspond to beta MNs, but that we refer to throughout the manuscript as "Type 3 MN."

Reviewer #2 (Remarks to the Author):

The revised manuscript has addressed all my concerns and is very much improved.

Reviewer #3 (Remarks to the Author):

In the revised version of the manuscript, the authors have substantially improved the quality of the manuscript and have addressed the main points of the reviewer. This refers to the 1) identification of specific novel markers for the individual MN subtypes, 2) the retrograde labelling of different muscles and 3) and improvement of the discussion with regard to the interpretation of their findings.

In summary, the present study represents an important basis for a deeper understanding of the diversity of cholinergic neurons and the reviewer has no further concerns.